# Home cage-based insights into motor learning and strategy adaptation in a Huntington disease mouse model

Daniel Ramandi[1,2], Marja D. Sepers[1], Zefang Wang[1], Brian Han[1],
Cameron L. Woodard[1¤], Timothy H. Murphy[1], Lynn A. Raymond[1]*

1 Department of Psychiatry, Djavad Mowafaghian Centre for Brain Health, Vancouver, British Columbia, Canada, 2 Graduate Program in Cell and Developmental Biology, University of British Columbia, Vancouver, British Columbia, Canada

¤Current address: Department of Neurology, University of California, San Francisco, San Francisco, California, United States of America
* lynn.raymond@ubc.ca

## Abstract

Huntington disease (HD) is a genetic neurodegenerative disorder characterized by progressive motor dysfunction, cognitive decline, and neuropsychiatric symptoms. Assessing early motor skill deficits in HD mouse models is challenging with traditional behavioral tasks. This study uses a home cage-based lever-pulling task, PiPaw2.0, to evaluate motor learning in 6–7 months-old zQ175 knock-in HD mice in a more naturalistic environment. In this task, mice learn to pull a lever for a water reward, with the requirement to hold the lever within a specific goal range for a required hold time. As the mice improved, the required hold time increased, thereby gradually increasing the task demands. Both wild type (WT) and zQ175 mice initially showed similar task engagement, but zQ175 mice had significant deficits in adapting to increasing hold time. The WT mice refined their strategies over time, shifting from random to more precise lever pulls, while zQ175 mice failed to make this adjustment, maintaining erratic performance. Additionally, in group-housing WT mouse lever performance benefited from peer interactions, an effect absent in zQ175 mice. Post-task neural assessments revealed that WT mice developed experience-mediated synaptic plasticity in the left striatum (contralateral to lever-pulling paw), while zQ175 mice showed no significant changes, consistent with known corticostriatal plasticity impairments in HD mouse models. In conclusion, our findings demonstrate the effectiveness of group-housed, home cage-based assessments for evaluating motor learning and adaptation in HD mouse models. This study provides insights into the motor control and adaptive learning deficits in HD, emphasizing the value of automated home cage systems in advancing neurodegenerative disease research and highlighting the importance of peer influences on performance.

## Introduction

Huntington disease (HD), a genetic neurodegenerative disorder characterized by progressive motor dysfunction, cognitive impairment, and neuropsychiatric symptoms, arises from

**Data availability statement:** All data files are available from the FRDR data repository (https://doi.org/10.20383/103.0869)

**Funding:** Funding was provided by the Canadian Institutes of Health Research PJT-178043 and Natural Sciences and Engineering Research Council AWD-021238 to L.A.R, Canadian Institutes of Health Research foundation grant FDN-143209 to T.H.M, University of British Columbia Faculty of Medicine Graduate Award #6442 to D.R, and Canadian Institutes of Health Research Canada Graduate Scholarship-Doctoral to C.L.W. The funders had no role in study design, data collection and analysis, decision to publish, or preparation of the manuscript.

**Competing interests:** The authors have declared that no competing interests exist.

a CAG repeat expansion in the huntingtin gene (*HTT*). The pathology predominantly affects the striatum and cortex, leading to motor symptoms such as chorea, bradykinesia, and rigidity, alongside early deficits in voluntary movement control and learning [1–3]. Mouse models of HD, particularly the zQ175 knock-in model [4], have been instrumental in mimicking the neural dysfunction observed in HD, offering insights into the disease's progression and symptomatology. Despite their genetic alignment with human HD, interpreting behavioral outcomes in these models is often challenging due to confounding factors such as heightened stress responses of mice, complicating the assessment of motor coordination and cognitive functions [5].

The use of operant tasks within group-housed mouse home cages represents a significant advancement in behavioral testing. By reducing animal handling and stress, this approach promotes more naturalistic learning and enables the gathering of comprehensive longitudinal data [6,7]. This is particularly pertinent for assessing forelimb motor functions, a critical aspect of HD symptomatology [8,9]. PiPaw, the lever-pulling-for-water automated home cage system [9], facilitates continuous, unbiased data collection, highlighting individual and social learning behaviors in a communal setting. Here, we describe a new version, PiPaw2.0; this adaptation includes dynamic task difficulty that is adjusted based on individual animal performance, providing a more tailored and challenging learning environment that evolves as the animals learn.

The zQ175 knock-in model [4] can develop early motor manifestations at six months of age [10], presenting a unique opportunity to study HD in its nascent stages. However, several studies have failed to find significant behavioral differences at the early stage of this model [4,11,12], underscoring the need for sensitive and robust behavioral assays. Our study employs the PiPaw2.0 system to evaluate forelimb motor learning in 6–7 months old zQ175 mice, aiming to detect differences in behavior and learning strategies at an early stage. This study extends previous work on the PiPaw system by introducing significant changes to the task design and using a different mouse model. The original PiPaw study [9] was done with Q175FDN mice, a model that is fully symptomatic by 9 months of age [13]. Moreover, unlike the earlier version, which used a narrow-range lever-pulling task that did not require a specific duration of hold-time, PiPaw2.0 employs an adaptive hold-time task where the required hold duration of the lever dynamically adjusts based on each mouse's performance, promoting sustained engagement and continuous learning over months. The updated system also features a more robust codebase, eliminating software errors and enabling reliable long-term data collection, with all annotated data publicly accessible for further analysis.

Our findings reveal that while HD mice perform a similar number of trials with overall comparable success rates to WT mice throughout the task, they exhibit difficulties in adapting to increasing task demands. Moreover, our study suggests a potential influence of peer interactions on performance, particularly among WT mice, highlighting the home cage paradigm's utility in studying complex social and environmental factors in neurological disorders.

## Results and discussion

### Task performance and motor learning

This study utilized a home cage-based forelimb skilled lever-pulling task, PiPaw2.0, where 6–7 month-old mice were housed in their familiar environment. This age represents an early stage where subtle motor deficits in zQ175 mice can be detected [4,10]. The mice had access to a specially designed chamber attached directly to their cage, in which they must pull a lever to obtain their water (Fig 1, S1 Video). This setup allowed continuous, naturalistic interaction with the task, minimizing stress and external interference. In Stage 1, any lever pull by the mouse was rewarded with a water drop, facilitating the task engagement and familiarity with the lever mechanism. Upon achieving a minimum of 100 lever pulls in a day, the mice

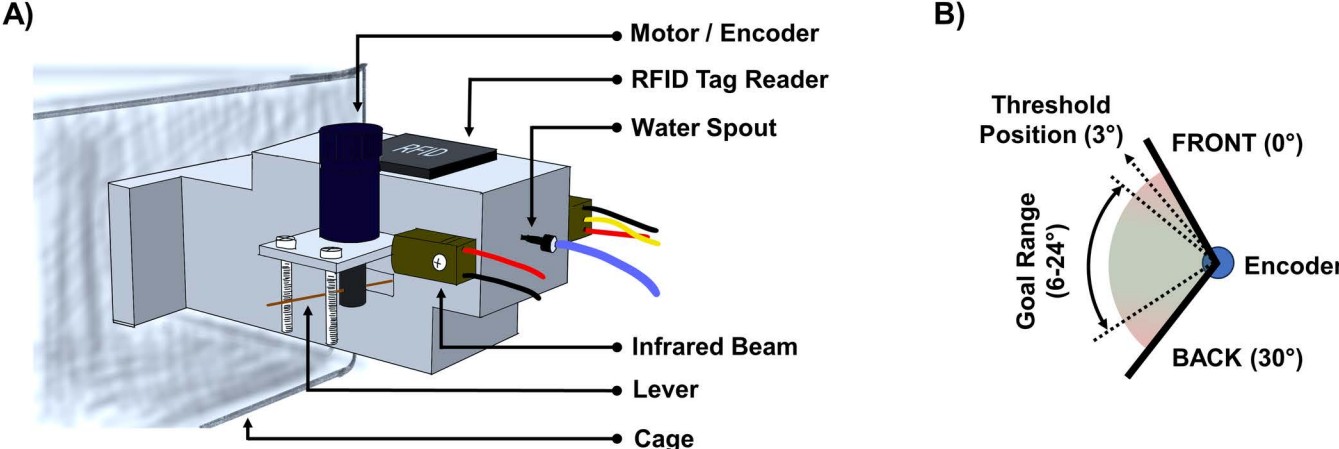

**Fig 1. Diagram of the PiPaw2.0 automated home cage lever-pulling system.** (A) The PiPaw2.0 system consists of a lever-pulling mechanism attached to the side of a mouse home cage. The setup includes the following components: a motor/encoder to measure lever position and movement while controlling the force applied to the lever, an RFID tag reader to identify individual mice, a water spout for dispensing water rewards, an infrared beam to detect nose pokes to start a trial, and the lever which mice must pull to receive a reward. (B) Schematic representation of the lever's movement range (0° to 30° from front to back) and the goal range for successful lever pulls. The threshold position for registering a pull is set at 3° to eliminate random lever movements and ensure a genuine pull. The goal range for a successful pull is between 6° and 24°. The duration that the lever must be held within this goal range is dynamically adjusted based on the performance of individual mouse to increase task difficulty and assess motor learning and execution.

were automatically advanced to Stage 2 the following day. This subsequent stage introduced a dynamic challenge, requiring mice to maintain the lever pull for a specific duration (referred to as hold time) to receive a reward. The required hold time in Stage 2 was adaptive, increasing in response to the mouse's performance improvements (refer to details in the methods section), thereby gradually increasing the difficulty of the task.

In assessing the initial engagement with the task, wild-type (WT) mice spent 7.6 ± 1.9 days on average in Stage 1, while zQ175 mice exhibited a longer duration of 11.3 ± 3.9 days (mean ± SEM). The difference between the groups was not statistically significant, suggesting comparable initial task engagement (Fig 2A).

Transitioning to Stage 2, the difficulty of the task was escalated progressively based on individual performance (see Fig 2B for representative traces of lever movement in Stage 2). Consequently, the daily success rate, which remained consistent at approximately 0.3–0.4 for both WT and zQ175 mice, did not serve as a reliable indicator of learning due to the adaptive difficulty adjustments (Fig 2C). A more revealing measure of learning was the average hold time of the lever. Initially, both WT and zQ175 mice improved in hold time over the first two weeks, indicating learning. However, while WT mice continued to extend their hold time over the subsequent weeks in the task, zQ175 mice reached a plateau at around 0.3 seconds, being unable to adapt to the heightened challenge of the task (Fig 2D).

We validated the results of the repeated measure two-way ANOVA on the daily average hold time by employing Intraclass Correlation Coefficient (ICC) and multilevel modeling (see methods section and Table A in S1 Appendix) [14,15].

Our findings align with several previous reports indicating operant learning deficits in various mouse models of HD [8,10,16–19]. These deficits primarily manifest in aspects of accuracy and reaction time, but not necessarily in initial task acquisition. This is corroborated by our findings, where both genotypes demonstrated similar engagement in the first stage of the task.

The most notable difference between WT and zQ175 mice appeared to be in their ability to adapt and change strategy in response to increasing task demands. This challenge in strategic

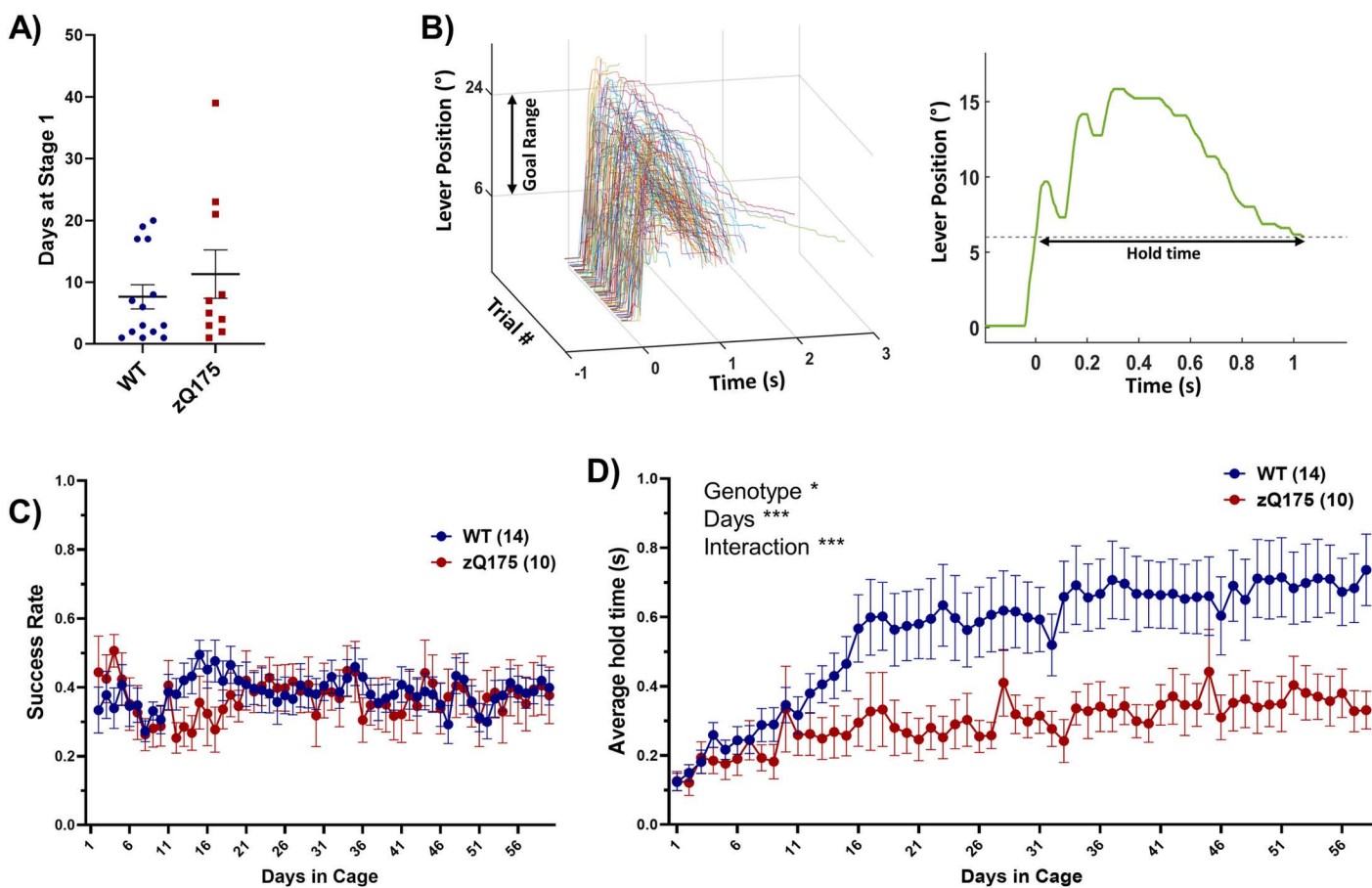

**Fig 2. Task performance and motor learning across genotypes.** (A) Analysis of the duration mice spent in the initial, less complex 'Stage 1' of the task revealed no significant difference between genotypes (Mann-Whitney test, p = 0.378). (B) Representative lever-pull trajectory plots from 100 consecutive trials of an expert WT mouse with a required hold time of 1 second (left panel). A detailed view of a single trial further illustrated the precision in hold time necessary to meet the task criteria (right panel). (C) Throughout Stage 2, daily success rates showed no significant variation between genotypes or across days in the cage (RM two-way ANOVA genotype p = 0.599 $F_{(1, 22)}$ = 0.2839, days p = 0.477 $F_{(8.331, 179.1)}$ = 0.9523, interaction p = 0.897 $F_{(58, 1247)}$ = 0.7696), suggesting a consistent performance level maintained by all mice. (D) The average daily hold time of WT mice showed an increase in response to the progressively demanding requirements of the task, while zQ175 mice reached a plateau at a significantly lower average hold time (RM two-way ANOVA, genotype p = 0.018 $F_{(1, 22)}$ = 6.432, days p < 0.0001 $F_{(4.369, 96.11)}$ = 9.192, interaction p < 0.0001 $F_{(57, 1254)}$ = 2.513). Plots in A, C, and D show mean±SEM.

adjustment in zQ175 mice could potentially be attributed to factors such as motor fatigue, motor coordination and control issues that are independent of fatigue, and/or motor perseveration. All of these factors have been observed in human HD patients, as demonstrated in hand-tapping tasks [20], as well as many measurements in the Q-Motor assessments [21]. This similarity in motor deficits highlights the potential value of our findings in providing insights into the motor control aspects of HD.

## Task engagement, activity, and motivation

To assess task engagement, we analyzed the number and pattern of trial performance. No significant difference in the number of daily trials between WT and zQ175 mice was observed, indicating similar levels of motivation and engagement (Fig 3A). Over the course of the study, both WT and zQ175 mice maintained consistent engagement with the task, as indicated by the stable number of daily trials performed. No significant reduction in task participation was

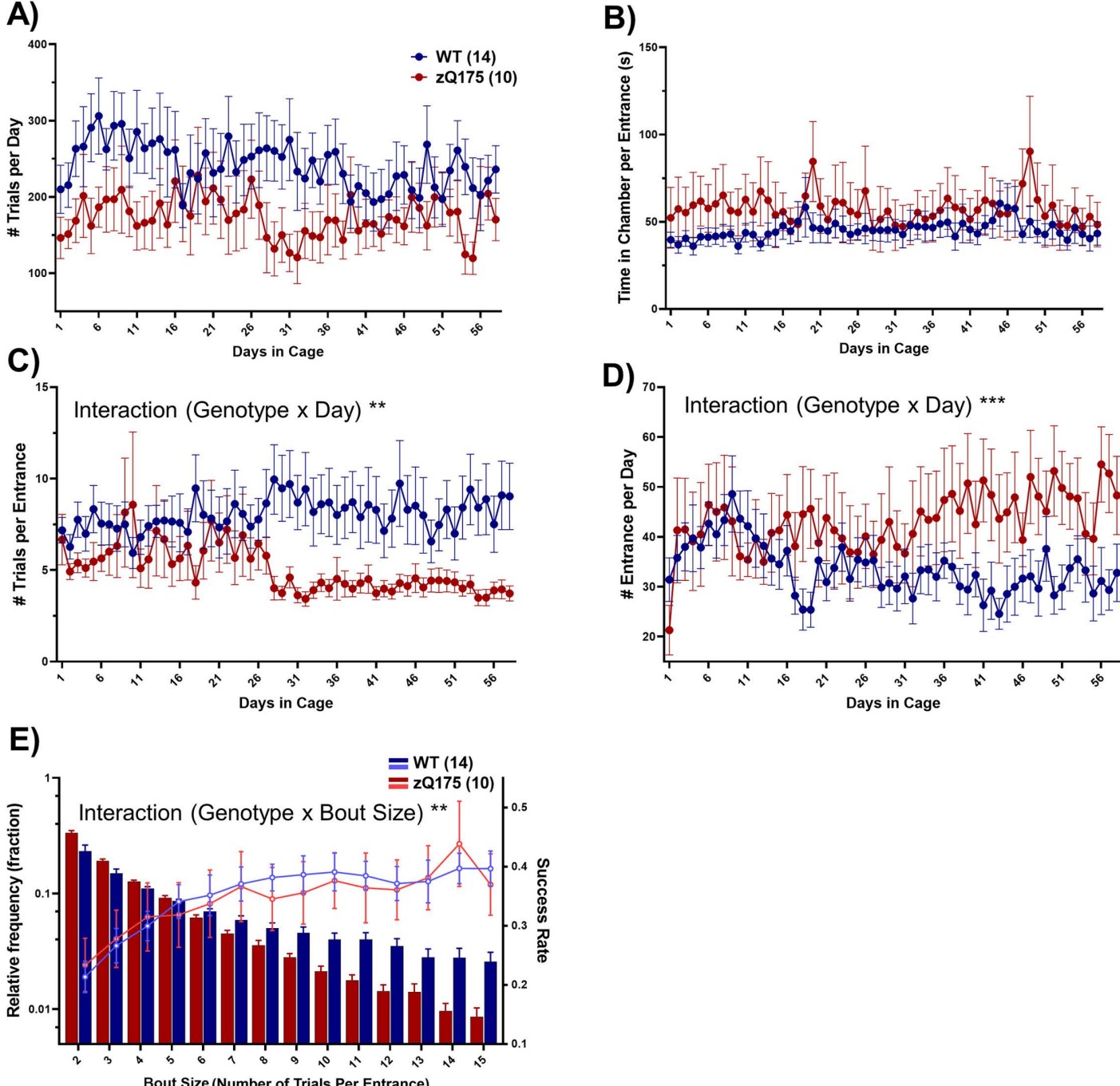

**Fig 3. Task engagement, activity, and motivation.** (A) Daily number of trials shows a nonsignificant trend toward a higher number of trials for WT mice compared to zQ175, suggesting similar levels of engagement between the two genotypes (RM two-way ANOVA, genotype p = 0.131 $F_{(1, 22)}$ = 2.420, days p = 0.535 $F_{(8.685, 191.1)}$ = 0.8867, interaction p = 0.789 $F_{(57, 1254)}$ = 0.8440). (B) The average time spent in the chamber per entrance showed no significant variations across days in cage or between genotypes (RM two-way ANOVA, genotype p = 0.390 $F_{(1, 22)}$ = 0.7684, days p = 0.290 $F_{(8.065, 175.9)}$ = 1.219, interaction p = 0.453 $F_{(8.065, 175.9)}$ = 1.219). (C) Analysis of the average number of trials per entrance revealed a divergence later in the testing period (after day 26), where WT mice performed more trials per entrance compared to zQ175 mice, hinting at a differential evolution in task engagement strategies between genotypes (RM two-way ANOVA over entire period of task engagement: genotype p = 0.065 $F_{(1, 22)}$ = 3.748, days p = 0.757 $F_{(6.148, 134.3)}$ = 0.5698, interaction p = 0.002 $F_{(57, 1245)}$ = 1.629). (D) Analysis of the frequency of chamber entries showed an opposite pattern to the average trial per entry, where WT mice entered less frequently after day 26 compared to zQ175 mice (RM two-way ANOVA over entire period of task engagement: genotype p = 0.183 $F_{(1, 22)}$ = 1.890, days p = 0.346 $F_{(8.782, 191.8)}$ = 1.126, interaction p < 0.0001 $F_{(57, 1245)}$ = 1.917). (E) The distribution of bout sizes (histogram; left y-axis log scale) and the corresponding average success rate for each bout size (line plot; right y-axis) uncovered a genotype-dependent difference in bout size distribution, with zQ175 mice showing a tendency for engaging in

fewer trials per bout. Additionally, success rates increased with bout size for both genotypes, yet no genotype difference in success rates was observed (RM two-way ANOVA, genotype $p = 0.245$ $F_{(1, 22)} = 1.426$, bout size $p < 0.0001$ $F_{(1.193, 26.24)} = 124.3$, interaction $p < 0.0001$ $F_{(13, 286)} = 6.476$ for distribution; genotype $p = 0.906$ $F_{(1, 22)} = 0.01428$, bout size $p < 0.0001$ $F_{(2.824, 62.13)} = 14.53$, interaction $p = 0.762$ $F_{(13, 286)} = 0.7004$ for success rate). Data in all plots are presented as mean±SEM.

observed, suggesting that neither genotype experienced task fatigue or motivational decline. Additionally, the average time spent in the training chamber per entrance did not differ significantly between the groups, although zQ175 mice displayed a non-significant trend towards longer durations (Fig 3B). Notably, the number of trials per chamber entry was consistent between WT and zQ175 mice for the first 25 days but subsequently diverged, with zQ175 mice performing fewer trials per entry (Fig 3C). The observed reduction in variability appears to arise from a decrease in variability between individual animals rather than a sudden behavioral shift. This stabilization likely reflects a gradual settling into more consistent engagement patterns across the group, rather than an abrupt change in task performance dynamics. This divergence, paired with an equal number of total daily trials as WT mice, suggested an increased frequency of chamber entries for zQ175 mice, which was confirmed (Fig 3D). Statistical analysis via two-way ANOVA revealed that the interaction of genotype and day was highly significant (Fig 3D), indicating a behavioral shift in zQ175 mice over time toward increased entrance frequency.

It is noteworthy that in line with previous reports on zQ175 mice, we observed a consistent difference in body weight between WT and zQ175 mice starting at approximately six months of age. While WT mice steadily gained weight over the course of the study, zQ175 mice exhibited minimal weight gain and remained 2–10% lighter on average (S1 Fig).

Previous studies have revealed that mice generally perform trials in concentrated bouts, as opposed to a random distribution [9]. Here, we defined a 'bout' as a sequence of consecutive trials initiated within one entry to the training chamber. The distribution of bout lengths across entries appeared to follow an exponential trend, with shorter bouts occurring more frequently than longer ones; however, WT mice were observed to engage in longer bouts more frequently compared to zQ175 mice (Fig 3E, histogram). Additionally, we investigated the average success rate corresponding to each bout size. The success rate was found to be higher for larger bout sizes and did not significantly differ between WT and zQ175 mice (Fig 3E, line plot). This suggests that while zQ175 mice engage less often in longer bouts of activity, their capability to perform, when they do, is not compromised. This observation partially rules out the involvement of motor fatigue and highlights the possibility of difficulties in sustaining proper motor control over extended periods, potentially intertwined with cognitive challenges like maintaining focus and attention [22,23]. Such patterns in zQ175 mice could reflect inherent aspects of HD that affect both motor and cognitive functions.

Interestingly, research in the field of motor learning suggests that activity patterns characterized by bouts of repetitive actions interspersed with pauses can be beneficial for motor skill learning [24]. These intermittent breaks are thought to facilitate the formation of classical Hebbian plasticity within the motor cortex [25], and are necessary for the modulation of motor variability [26]. For zQ175 mice, the reduced frequency of high-activity bouts could alter their ability to explore the task environment more thoroughly and learn through trial and error.

## Modulation of performance during learning

To explore how motor learning differs between WT and zQ175 mice, we analyzed the distribution of hold times during the task, expecting distinct patterns between random and precise trials. This approach aimed to reveal how each group modulates trial performance, with a focus

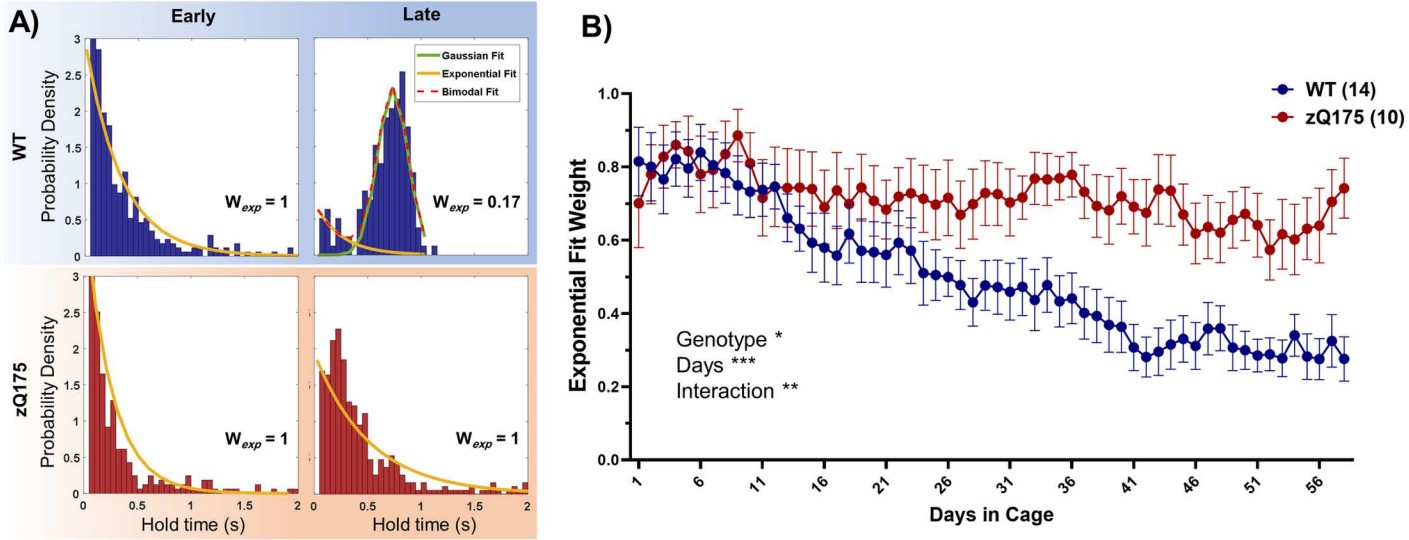

**Fig 4. Modulation of performance during learning.** (A) The distribution of daily hold times for a representative WT (top) and zQ175 (bottom) mouse during early (days 1–3) and late (days 56–58) stages in the cage, complemented by a bimodal fit analysis. Only the late stage in the WT mouse shows a Gaussian component (green line), indicative of more timed responses, a feature absent in their early performance and persistently missing in zQ175 mice at the late stage, whose distribution remains predominantly exponential. The '$W_{exp}$' value shows the weight of the exponential component in each distribution. (B) A longitudinal comparison of the average exponential fit weight between WT and zQ175 mice reveals that while WT mice show a decreasing trend, indicating a shift towards longer duration ("timed") trials, zQ175 mice maintain a consistently high exponential weight, demonstrating no significant change over time (b; RM two-way ANOVA, genotype p = 0.0106 $F_{(1, 22)}$ = 7.795, days p < 0.0001 $F_{(5.262, 115.4)}$ = 7.616, interaction p < 0.0001 $F_{(57, 1250)}$ = 2.834). Data presented as mean±SEM.

on understanding whether HD impairs the ability to transition from exploratory, random responses to more deliberate, precise actions during motor learning. We utilized a generative model to analyze hold time data (see methods section), similar to those that have been previously used [27,28]. During the learning process (first 2 days of training), an analysis of trial hold time distributions for both WT and zQ175 mice displayed an initial exponential trend (Fig 4A), indicating that trials were predominantly brief. This pattern is typical of the early learning phase, where interactions with the task are more exploratory than precise [29]. As learning advanced, WT mice showed a significant transition towards a Gaussian distribution of hold times centered around the task's required duration (Fig 4A, B). This evolution in the distribution pattern underscores a strategic refinement in performance, where WT mice adjusted their behavior to achieve the desired hold times. Contrastingly, zQ175 mice maintained a hold time distribution that was largely exponential with a skew towards brief hold times, without a marked emergence of a Gaussian profile (Fig 4A, B). This persistent pattern suggests a less effective modulation of trial performance, despite the fact that the average daily success rate remained constant and comparable to that of WT mice throughout the learning period. This emphasizes that the primary difference lies in the modulation of trial performance—how the mice strategize their trials—rather than in their ability to achieve success per se.

The fast, seemingly short trials may reflect random performance of the task. Drawing from decision modeling literature [30], we categorize these as "untimed" movements, which lack a deliberate temporal structure, in contrast to "timed" movements that conform to specific criteria and exhibit deliberate temporal control. The exponential distribution of these untimed trials indicates that shorter hold times are more frequent than longer ones. This pattern contrasts with a random uniform distribution, where all hold times between 0 and 1 second are equally probable. The exponential distribution suggests that these trials are fundamentally random in nature, but the observed distribution is shaped by the energy requirements of the

task—longer hold times, which demand more energy, are less likely to occur. Thus, while the trials are random, their distribution reflects an interaction with energetic constraints.

Despite not aligning with the optimal strategy as defined by our task, these untimed responses persisted even among the expert WT mice. One explanation is that those persistent short-duration responses reflect periods of frustration, fatigue or random slips; alternatively, they may serve a purpose beyond mere random task failure. One plausible interpretation is that these are exploratory responses. In this view, quick, low-cost responses may be a strategic choice to minimize the potential loss from not promptly adapting to environmental changes, as suggested by previous research [31]. Such a strategy might be particularly relevant in dynamic environments where conditions can shift unexpectedly.

However, in zQ175 mice, the minimal shift away from this exponential distribution suggests a different dynamic. It raises the possibility that these short-duration responses might be impulsive and less adaptive, or simply a manifestation of a tendency to engage in stereotypical response patterns across varying task settings [32]. The observed initial increase in hold time among zQ175 mice is reflected in a decrease in the decay rate (lambda) of the exponential distribution. This suggests that instead of developing a Gaussian distribution centered around the required hold time, as seen in WT mice, zQ175 mice extend their hold times by prolonging the tail of the exponential distribution (widening the distribution). This strategy results in a more energy-intensive and suboptimal task performance, as it involves a higher frequency of untimed, brief trials rather than a strategic adjustment toward the required duration. Consequently, zQ175 mice demonstrate a limited capacity to adapt their hold times efficiently, likely due to the cumulative energetic cost and inefficiency of their approach. This insight into the response dynamics of zQ175 mice contributes to a deeper understanding of how HD may impact both motor control and decision-making strategies in response to task demands.

## Kinematic analysis: Jerkiness of movement

An integral part of our analysis involved examining the kinematics of lever movement, specifically focusing on its 'jerkiness'. To assess this, we applied a high-pass filter with a cutoff frequency of 10 Hz to the lever trajectory data, as previously done in literature to measure and quantify tremor [33,34]. The rationale behind this filtering approach is that it isolates rapid fluctuations in the movement signal, effectively capturing the high-frequency components that constitute jerky movements. Following this filtering, we computed the standard deviation of the filtered signal, providing a quantitative measure of movement jerkiness (Fig 5A).

Upon analyzing the daily averages of the jerkiness metric, a notable distinction emerged between the WT and zQ175 mice. While both groups exhibited similar levels of jerkiness during the initial 10 days, the zQ175 mice demonstrated a significantly higher jerkiness in their movements as the task progressed (Fig 5B). This was evident not only in the overall average (WT: 0.077 vs. zQ175: 0.098, p = 0.017) but also in the repeated measure daily averages (shown in Fig 5B), suggesting a consistent genotype-specific difference.

The heightened jerkiness observed in zQ175 mice indicates greater variability and less smoothness in their high-frequency movements during trials. Remarkably, WT mice maintain a low level of jerkiness even as their average hold duration increases. This contrast becomes more pronounced after the first month in the cage, where zQ175 mice show a significant increase in jerkiness despite their average hold time remaining relatively unchanged. This might be due to motor deficits in zQ175 mice becoming progressively more pronounced over time. Similar motor changes have been noted in patients with pre-symptomatic HD, exhibiting reaching movements characterized by increased jerkiness and impaired movement termination [3]. Further, in a water-reaching task, we have recently demonstrated a similar

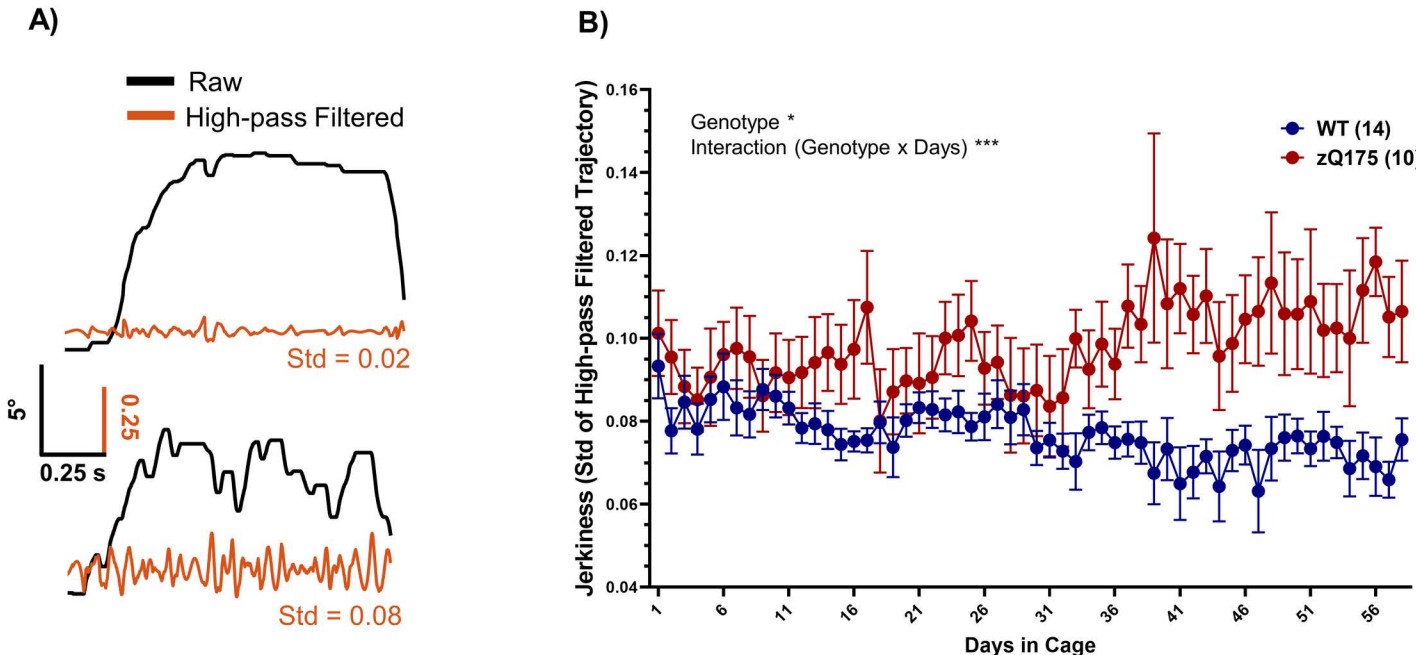

**Fig 5. Analysis of movement jerkiness.** (A) Two example traces that illustrate a smooth (top) versus jerky (bottom) lever pull trajectory (black traces) with the corresponding 10Hz high-pass filtered trajectories superimposed (orange traces). The standard deviation (Std) of these high-pass filtered trajectories quantitatively captures the movement's jerkiness. (B) The daily averages of Std for the high-pass filtered trajectories across WT and zQ175 mice further reveals significant differences in movement jerkiness, with zQ175 mice consistently exhibiting higher levels of jerkiness compared to WT mice, a difference that becomes more pronounced later in the testing period as average hold times increase (RM two-way ANOVA, genotype p = 0.017 $F_{(1, 22)}$ = 6.576, days p = 0.463 $F_{(8.019, 176.4)}$ = 0.9676, interaction p < 0.0001 $F_{(57, 1254)}$ = 2.762). Data presented as mean ± SEM.

phenotype in reach trajectories in early manifest zQ175 mice [10], providing a parallel to human HD manifestations in a reach-to-eat task [35].

The ability to measure such movement qualities in our automated platform is particularly noteworthy. This system facilitates the collection of extensive data on movement in an efficient and precise manner, contrasting sharply with the more labor-intensive and less scalable markerless video tracking strategies typically employed to analyze fine movements.

## Variability in performance and response to changing task requirements

Our analysis of behavioral variability employed two main measures: dynamic time warping (DTW) to gauge the Euclidean distance between lever trajectories of consecutive trials (as a measure of dissimilarity between successive trials) and a moving standard deviation of hold times over five-trial windows (as another measure of trial-to-trial variability). These measures were chosen to accommodate the varying lengths and temporal structures of the trials, offering a deeper analysis of performance dynamics.

**Response to hold time changes:** Initially, we examined how mice adapted their trial-to-trial variability in response to changes in the required hold time. Two zQ175 mice were excluded from this analysis as they remained at the same hold time in Stage 2 and did not meet the criteria for increasing the initial hold time. Focusing on the period encompassing five trials before and fifteen trials after a change in required hold time, there was a significant reduction in the success rate for both WT and zQ175 mice following the change in hold time, indicating an immediate impact of heightened task difficulty (Fig 6A). This decrease was consistent across genotypes. We also observed an overall increase in trial-to-trial variability in

lever trajectory, as measured by DTW, shortly after the new hold time was introduced (Fig 6B). This suggests an adjustment period where mice recalibrate their strategies to the new task demand. A trend was noted towards greater variability in zQ175 mice, taking them longer to adapt their performance, although this did not reach statistical significance.

Concurrent with these changes, we observed a rise in the variability of hold times, as indicated by the moving standard deviation of hold times, which mirrored the increase in trial-to-trial variability in lever trajectory (Fig 6C). This heightened variability suggests that both WT

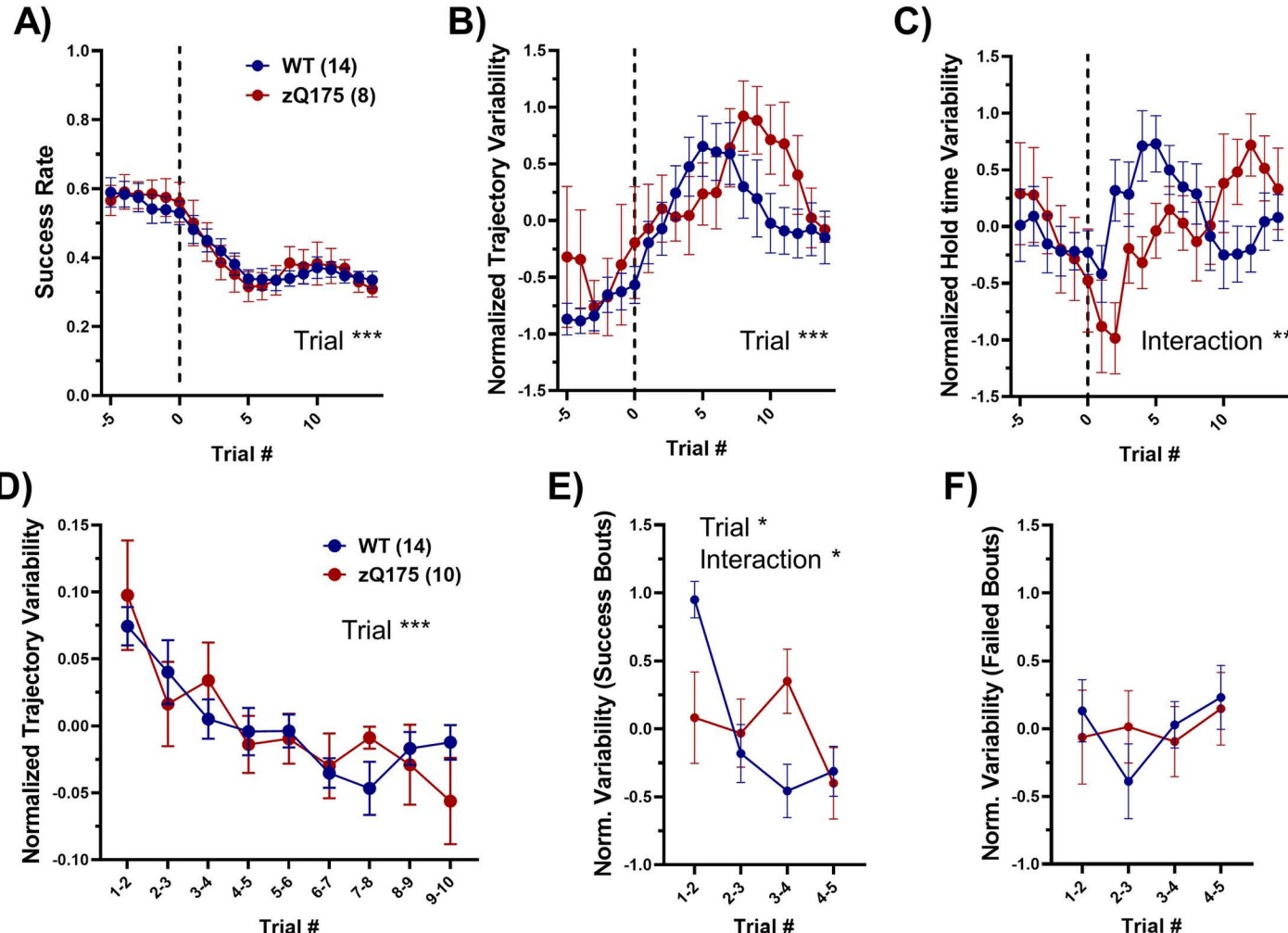

**Fig 6. Variability in performance and response to task dynamics.** (A) Upon a change in the required hold time (dashed line), a significant reduction in success rate was similarly observed for both WT and zQ175 mice (RM two-way ANOVA, genotype p = 0.926 F(1, 20) = 0.008736, trial p < 0.0001 F(3.915, 78.31) = 27.99, interaction p = 0.985 F(19, 380) = 0.4237). (B) The analysis of trajectory variability, quantified as DTW distance of consecutive trials, showed a non-significant trend towards greater variability in zQ175 mice, peaking later compared to WT mice (RM two-way ANOVA, genotype p = 0.070 F(1, 20) = 3.660, trial p = 0.0005 F(4.291, 85.82) = 5.351, interaction p = 0.542 F(19, 380) = 0.9329). (C) Similarly, hold time variability, measured by moving standard deviation of hold times, presented a pattern where zQ175 mice exhibited a later peak in variability, suggesting a slower adaptation process (RM two-way ANOVA, genotype p = 0.571 F(1, 20) = 0.3293, trial p = 0.155 F(4.597, 91.94) = 1.668, interaction p = 0.0075 F(19, 380) = 2.011). (D) Further examination of trajectory variability within a 10-trial bout showed no differences between genotypes, with both demonstrating a reduction in variability (RM two-way ANOVA, genotype p = 0.630 F(1, 22) = 0.2383, trial p = 0.0002 F(4.154, 91.39) = 6.081, interaction p = 0.693 F(8, 176) = 0.6970). (E) When focusing on five consecutive successful trials, WT mice showed a significant reduction in trajectory variability, whereas zQ175 mice exhibited only a slight decrease, predominantly in the last trial, hinting at genotype-specific differences in optimizing performance following success (RM two-way ANOVA, genotype p = 0.746 F(1, 22) = 0.1074, trial p = 0.012 F(2.836, 62.38) = 4.045, interaction p = 0.018 F(3, 66) = 3.561). (F) Conversely, in five consecutive failed trials, no significant change in trajectory variability was observed for either genotype (RM two-way ANOVA, genotype p = 0.755 F(1, 22) = 0.09948, trial p = 0.631 F(2.664, 58.61) = 0.5467, interaction p = 0.743 F(3, 66) = 0.4142). Data presented as mean±SEM.

and zQ175 mice were exploring a broader range of movements in the action space to meet the new requirements. Notably, the peak in variability was delayed in zQ175 mice compared to WTs, indicating a slower adaptation phase following the task alteration. This delay, akin to the trend observed in trial-to-trial variability, further exemplifies the zQ175 mice's extended adjustment period to new task demands.

**Variability in extended trial bouts:** Further, we investigated the first 10 trials in trial bouts consisting of 10 or more trials within a 5-minute window. Consistent with existing literature [9,36], we found a general decrease in variability of lever trajectories across such extended bouts (Fig 6D), indicating a settling into a more stable performance pattern as the task continued. This trend was uniform across both WT and zQ175 mice, suggesting that prolonged engagement with the task facilitates a more consistent approach in both genotypes.

**Outcome-based variability patterns:** In a more focused analysis, we looked at sequences of five consecutive trials, either all successful or all failed, occurring within a 150-second timeframe. WT mice exhibited a significant reduction in movement variability during successful trial bouts (Fig 6E), primarily during the first three trials, suggesting a strategic refinement based on positive feedback. Conversely, zQ175 mice showed minimal changes in variability, with a slight reduction only in the last trial of the bouts. This absence of a clear trend in zQ175 mice, particularly in the face of success, implies a deficit in reward-modulated behavioral adjustment. Surprisingly, failed trials did not mirror this pattern for either genotype, indicating that success, rather than failure, plays a more pivotal role in modulating performance strategies in this task (Fig 6F).

The consistent daily success rates between WT and zQ175 mice (Fig 2C), juxtaposed with the differences in their response to task dynamics and feedback, paint a complex picture of motor learning and adaptation. The zQ175 mice's slower adjustment to changes and less pronounced variability reduction in response to success suggest a possible deficit in integrating performance feedback or in the cognitive flexibility required for task adaptation. This might reflect underlying differences in neural circuitry or reward-modulated synaptic plasticity between the genotypes. Further, the lack of significant variability change in response to failure could indicate that positive reinforcement is a more potent driver of strategy optimization in both genotypes in this motor learning paradigm. These insights hint at the nuanced interplay between motor control, cognitive processing, and feedback sensitivity that underlies adaptive learning in complex tasks.

## Inter-mouse influence in a group-housed automated home cage setting

In exploring the potential influences of group housing on learning, we examined specific activity motifs within each cage (Fig 7A and B). For a pair of mice in a cage these motifs involved a sequence where one mouse (the "follower") completed at least 10 trials within 5 minutes, followed by a second mouse (the "influencer") performing a similar number of trials within the same time frame, and then the follower mouse returning for another bout of trials (Fig 7A). The interval between the follower and the influencer's activities was under 5 minutes, ensuring relevance in their sequential engagement. The observed frequency of these interaction patterns was consistent across genotypes, occurring on average 156 ± 27 times (mean ± SEM) over a two-month period, which, while not highly frequent (approximately two to three times daily), is substantial given the large dataset and the stable average daily success rate observed during this period (Fig 7C).

Interestingly, when the influencer mouse outperformed the follower in their middle trial bout (referred to as good influencer), the follower exhibited an improved performance in its subsequent bout (Fig 7D). This enhancement was not observed if the influencer mouse performed worse than the follower (referred to as bad influencer) (Fig 7E). To control for

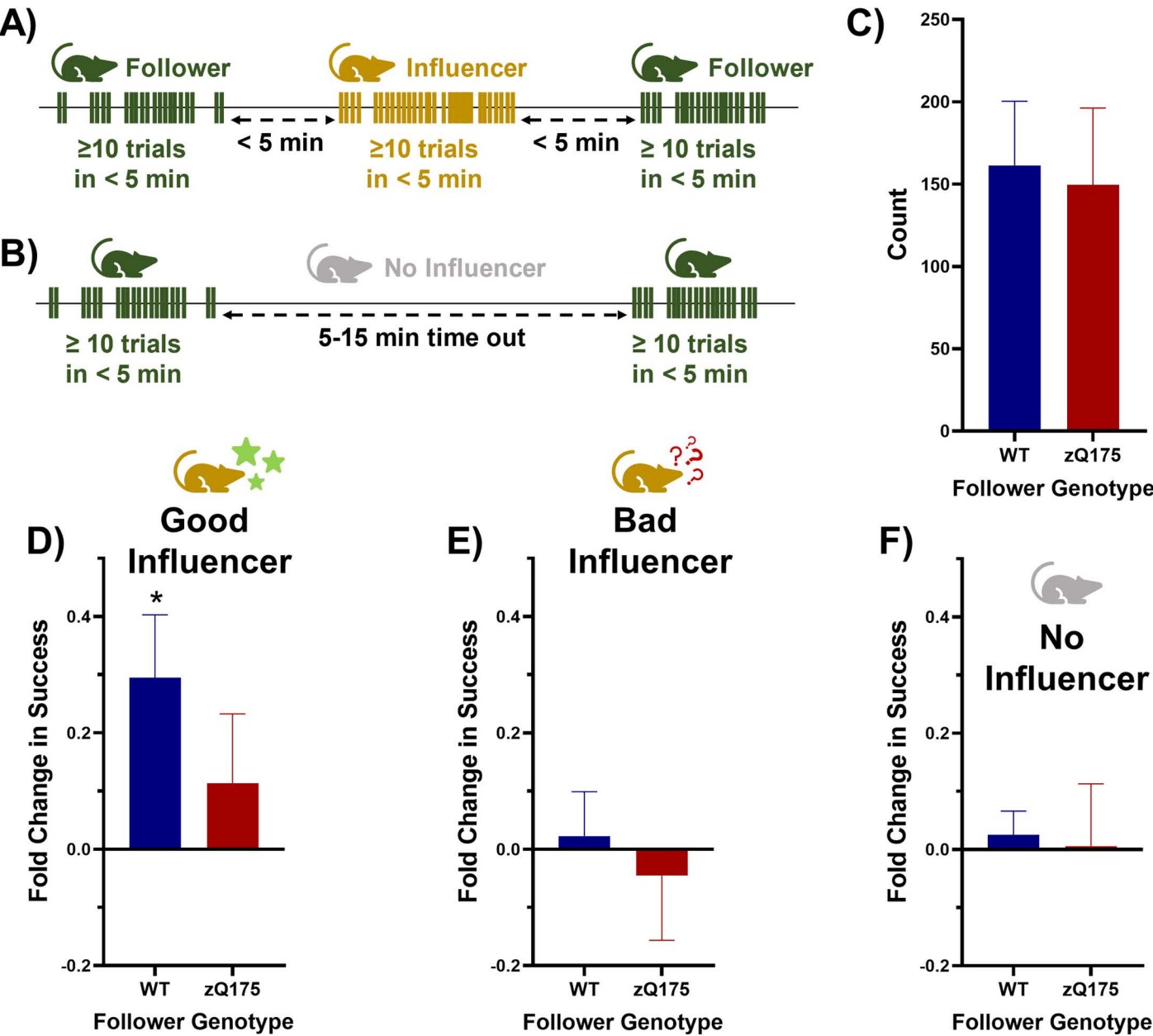

**Fig 7. Inter-mouse influence in a group-housed automated home cage setting.** (A) diagrammatic representation of the interaction motif between pairs termed as follower-influencer, specifying the criteria for identifying these interactions. (B) For comparative analysis, control patterns devoid of an influencer presence were also examined, characterized by a solitary mouse performing two bouts separated by a 5–15 minute timeout. (C) The frequency of follower-influencer motifs identified throughout the study period revealed no significant difference in occurrence based on the genotype of the follower, indicating similar levels of this interactive behavior across genotypes (unpaired t-test, p = 0.850). (D) In cases when an influencer exhibited higher success rates than the follower's baseline (good influencer), WT but not zQ175 followers displayed a significant increase in success rate (one-sample t-test, theoretical mean 0.0, WT p = 0.014, zQ175 p = 0.366), although no genotype differences were noted when comparing these improvements (unpaired t-test, WT vs. zQ175 p = 0.290). (E) Conversely, when influencers had the same or lower success rates compared to the follower's baseline (bad influencer), both WT and zQ175 followers showed no significant deviation from their initial success rates (one-sample t-test, theoretical mean 0.0, WT p = 0.766, zQ175 p = 0.695; unpaired t-test, WT vs. zQ175 p = 0.615). (F) The control scenarios, lacking an influencer and merely documenting the variance in success rates across two separated bouts, mirrored the outcomes observed with bad influencers, with no significant changes in success rates for either genotype (one-sample t-test, theoretical mean 0.0, WT p = 0.561, zQ175 p = 0.957; unpaired t-test, WT vs. zQ175 p = 0.851). Data presented as mean±SEM.

intrinsic performance fluctuations, we also analyzed patterns where a single mouse completed two separate 10-trial bouts with a 5–15 minute interval (Fig 7B), finding no significant change in performance in these solitary bouts (Fig 7F).

This pattern, intriguingly, was prominent only when the follower was a WT mouse. When a zQ175 mouse acted as the follower, its performance did not significantly change regardless of the influencer's success. This suggests a genotype-specific response to the perceived success of a peer, potentially influenced by auditory cues (success and failure tones after each lever-pulling trial) since direct visual observation was not possible in our setup.

These findings, while preliminary, highlight the intriguing potential of group-housed automated home cage experiments in uncovering peer influences on learning and performance. The observed 'peer-influenced performance adjustment' suggests a form of indirect learning or motivation modulation, where the success of a peer, inferred possibly through auditory cues, enhances subsequent performance in WT mice. This phenomenon could reflect a heightened sensitivity in WT mice to the performance cues of their peers, a trait seemingly less pronounced in zQ175 mice. This differential response might be rooted in genotype-specific variations in social cognition or motivational factors. The lack of a similar response in zQ175 mice could indicate a diminished capacity to utilize peer performance as a motivational or learning cue, which aligns with some of their previously observed learning challenges.

The utility of automated home cages in this context is particularly noteworthy. Such systems enable the long-term, detailed tracking of individual behaviors in a group setting, providing a unique lens through which complex social interactions and their impact on learning can be studied. This methodological approach could be pivotal in unraveling the subtleties of social learning dynamics, particularly in genetically diverse populations.

## Genotypic and hemispheric differences in striatal plasticity

In the final segment of our study, we focused on assessing neural plasticity following a 2–3 month period of behavioral tasks in the automated home cages. Mice were euthanized, and field excitatory postsynaptic potentials (fEPSP) were recorded from the dorsolateral striatum in sagittal slices (Fig 8A, B) The high-frequency stimulation (HFS) protocol was applied in the presence of picrotoxin to block GABAAR (see methods section).

Our analysis revealed significant genotypic and hemispheric variations in plasticity induction (Fig 8C, D). Notably, in WT mice, we observed long-term potentiation (LTP) in the left hemisphere (contralateral to the lever-pulling forepaw; Fig 8D). There was no significant hemispheric difference or plasticity induced by HFS in the zQ175 mice.

The lack of change in synaptic response in zQ175 is consistent with studies showing impaired corticostriatal plasticity in HD mice that have had no behavioural testing [37]. Similarly, we have previously shown that spontaneous activity at corticostriatal synapses in naive WT mice is changed by experience with a precision lever task [9]. In that study, spontaneous excitatory events (reflecting basal glutamatergic synaptic activity) in the left hemisphere of WT mice after lever-pulling with the right forepaw showed a decrease in amplitude that correlated with success, but were unchanged in HD mice that were unable to learn the task. The hemispheric difference in synaptic plasticity shown here in WT striatum suggests an *in vivo* change due to learning a motor task that increases in difficulty. It is interesting to note that the zQ175 did not show synaptic plasticity and were unable to reach the level of performance of WT mice. Previous studies have also shown differences in striatal plasticity after rotarod or t-maze learning [38,39].

One notable observation is the absence of LTD in the right hemisphere of WT mice, which typically manifests in naive animals following HFS in the dorsolateral striatum [37]. A potential explanation for this deviation could be the cognitive enrichment and extensive experience

provided by the automated task chamber (not necessarily the unilateral motor learning). This aligns with the idea that enriched environments can modulate synaptic plasticity, potentially stabilizing synapses and/or occluding further *ex vivo* HFS-induced plasticity, thereby preventing the typical LTD response [40].

The LTP observed in the left hemisphere of WT mice could be interpreted through the lens of metaplasticity [41]. One hypothesis is that the reduced basal sEPSC amplitude in the WT left hemisphere, reflecting chronic LTD from motor learning, creates a condition where the HFS protocol induces LTP. According to the Bienenstock-Cooper-Munro (BCM) theory [42], which posits a sliding modification threshold for synaptic plasticity, the threshold for LTP induction decreases when previous synaptic activity has been low. In this context, the ongoing motor learning could have lowered the baseline synaptic activity in the left hemisphere [9], making it more susceptible to LTP when subjected to HFS. Although this theory is well-documented in the hippocampus, and visual and motor cortices [41], its application to striatal plasticity is still hypothetical. Nevertheless, the presence of a hemispheric difference in

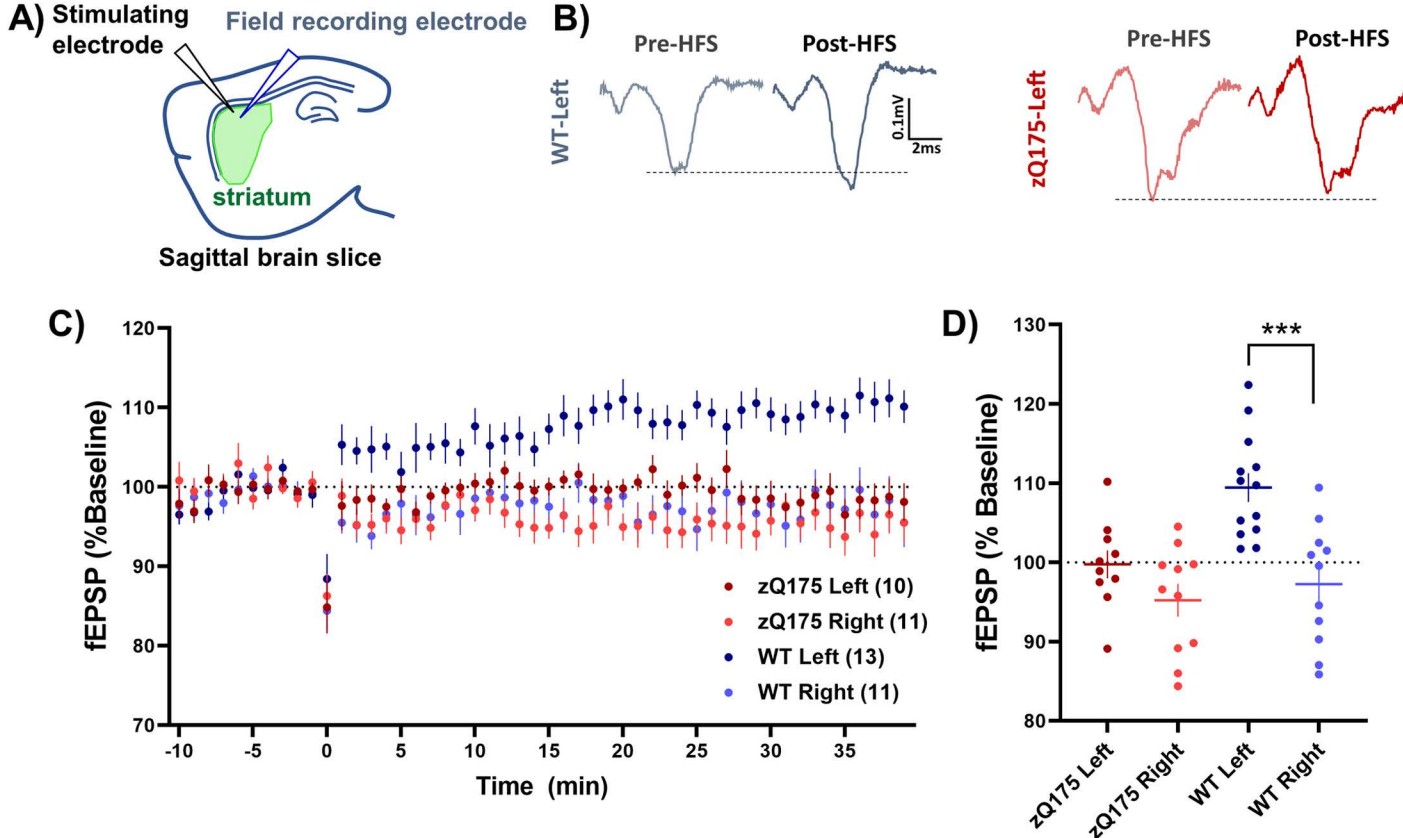

**Fig 8. Genotypic and hemispheric differences in striatal plasticity post PiPaw2.0.** (A) Schematic representation outlining the experimental setup, detailing the placement of stimulating and recording electrodes within the dorsolateral striatum. (B) Representative field responses before and after plasticity induction through high-frequency stimulation (HFS) depicted for both WT and zQ175 mice in the left hemisphere. (C) The dynamic changes in response amplitude relative to the baseline, following HFS, illustrated over the course of the plasticity experiment, providing a temporal view of the neural responses in WT and zQ175 mice, in the left hemisphere (contralateral to the lever-pulling paw) and right hemisphere (ipsilateral to the lever-pulling paw). (D) An analysis of the change in response size, averaged over minutes 30–35 post-HFS, revealed a significant hemispheric difference in plasticity induction among WT mice, a difference not observed in zQ175 mice (Two-way ANOVA, genotype p = 0.005 $F_{(1, 41)}$ = 8.493, hemisphere p = 0.0002 $F_{(1, 41)}$ = 17.23, interaction p = 0.064 $F_{(1, 41)}$ = 3.605; Sidak's multiple comparison left vs. right, WT p = 0.0001, zQ175 p = 0.243). The numbers in parentheses in C and the points in D reflect number of slices from a total of 7 WT (4 male) and 5 zQ175 (3 male) mice. Data presented as mean±SEM.

synaptic plasticity in WT mice underscores the influence of experience-mediated plasticity, as evidenced by the differential response between the left and right hemispheres, and as observed in our previous study [9].

While our findings contribute to the study of *ex vivo* plasticity after *in vivo* experience, the mechanism and direct linkage to the specific motor experiences of the mice in our study remains undetermined.

## Conclusion

Our study underscores the exceptional utility of automated, group-housed home cage tasks in behavioral research. This setup enables the collection of extensive, unbiased data over prolonged periods mitigating the impact of behavioral stochasticity and offering a more accurate representation of naturalistic behaviors.

In the context of HD, our findings revealed that while HD mice perform a similar number of trials with comparable success rates to their WT littermates, they are unable to meet escalating task demands. This challenge manifests in two key aspects: a reduced frequency of high-activity bouts and an inability to modulate activity effectively, notably reflected in their tendency for shorter, impulsive trial attempts contributing disproportionately to their overall performance distribution. Furthermore, our results suggest that animals within these group-housed settings may influence each other's performance, with WT mice showing a more pronounced response to peer behavior. This finding not only speaks to the dynamic nature of learning and adaptation in a communal environment but also showcases the potential of the home cage paradigm to unravel complex social learning and interaction patterns.

While these observations provide valuable insights into early HD-related phenotypes, further studies incorporating additional biological measures will be essential to confirm causative mechanisms and fully elucidate the link between functional deficits and underlying neuropathology. This study, therefore, positions the automated home cage systems as a powerful tool for advancing our understanding of animal behavior, particularly in the context of motor disorders like HD.

## Materials and methods

### Animals

All experimental protocols were conducted per the ethical standards set by the Canadian Council on Animal Care (CCAC) and received full approval from the University of British Columbia Committee on Animal Care (protocol A23-0083). This study utilized male and female heterozygous neo-deleted knock-in zQ175 Huntington Disease (HD) mouse models (B6.129S1-Htttm1.1Mfc/190ChdiJ; Jax Stock No. 029928) aged 6–7 months at the start of the lever-pulling-for-water task. C57BL/6J wild-type (WT) littermates, lacking the expanded allele, served as controls. Genotyping was performed by PCR on tissue collected via ear clipping at weaning; only zQ175 mice with a CAG repeat length within the range of 190–210 were included in the study. A total of 8 female (4 WT and 4 zQ175) and 16 male (10 WT and 6 zQ175) mice were used for experiments described here. Our group size was informed by prior studies that detected significant motor learning differences in HD models using similar automated home-cage paradigms [8,9]. Mice were housed at a density of 2 or 3 mice per cage, and including either or both genotypes, as genotypes were not separated.

The mice were housed under a controlled 12/12-hour light/dark cycle with regulated temperature and humidity conditions. Standard environmental enrichments, including bedding (1/8" pelleted cellulose), a hut, and wooden chew toys, were consistently provided within their cages to promote well-being before and throughout the behavioral testing period.

For the facilitation of automated identification of group-housed mice and behavioral analysis, glass Radio-Frequency Identification (RFID) capsules (Sparkfun SEN-09416) were subcutaneously implanted in the back of the neck of each mouse.

Animal weights were monitored thrice weekly throughout the testing period to ensure adequate food and water intake. A protocol was established whereby any mouse experiencing a weight reduction below 20% of their pre-testing baseline would be immediately removed from the study and placed back into conventional housing with unrestricted access to water and food to facilitate recovery, and put back to the automated cage (a total of 5 instances, due to software failure). In all five instances, the mice fully recovered within 1–2 days and resumed their previous activity patterns and task performance upon reintroduction to the automated home cage, with no lasting signs of distress or dehydration, though such events could introduce minor confounds in the data. During the period of experiments, 2 female mice (1 WT and 1 zQ175) housed together were removed from the automated cage, and excluded from analysis, due to lack of any engagement with the testing chamber.

## Hardware and software integration

The PiPaw2.0 system, an enhancement of the original PiPaw home cage behavioral testing framework [9], was designed for advanced behavioral experiments. A standard mouse home cage (7.5" W x 11.5" L x 5" H) was modified by incorporating an access portal to a 3D-printed training chamber. Within this chamber, a nose-poke port with a water spout was positioned at the far end of the entrance, equipped with a gravity-operated solenoid valve system to deliver 10uL water drops. Overhead, an RFID antenna and reader (Sparkfun SEN-11 828) were embedded to identify and log the mice via their subcutaneous RFID tags.

Adjacent to the water spout, a horizontally pivotable lever projected 1.5 cm into the chamber, allowing for 30° of movement, corresponding to approximately 1 cm at the tip. This lever was strategically placed to engage the mouse's right forelimb naturally during water retrieval attempts. Directly across, a supportive ledge was provided for the left forelimb, facilitating balance during the simultaneous lever manipulation and nose-poking activities.

The lever itself was connected to a direct current (DC) micro-motor (Faulhaber 1524T012SR) which maintained the lever in a default 'start' position under specific torque conditions. This system employed dual-force settings: a minimal 'low-force' during active trials to facilitate motion with roughly 15 mN of force, and a 'high-force' mode to secure the lever's position during intertrial intervals. The integrated high-resolution encoder (Faulhaber IEH2-4096) allowed for precise monitoring and documentation of lever angular position at a sampling rate of 400 Hz.

To observe the behavior, a camera (Waveshare 10299) was mounted beneath the module, capturing trial events from below through a transparent floor. Auditory cues for trial start, reward, and non-reward events were generated by a piezo buzzer, mounted on the external side of the training module. All electronic components interfaced with a custom-designed printed circuit board (PCB), which, in conjunction with a Raspberry Pi 3B micro-computer, facilitated the operation and synchronization of the system. Custom-developed Python software was utilized for system control, with the source code and operational guidelines made publicly available (https://github.com/ubcbraincircuits/PiPaw).

## The PiPaw2.0 task

The behavioral task designed for this study was a 'hold task', requiring mice to maintain lever position within a predefined range for a set duration. Upon entry into the chamber, each mouse's presence was detected by the RFID system, triggering the loading of its specific data

profile. Initiation of a trial commenced with a nose-poke by the mouse, disrupting an infrared beam. This was marked by a medium-pitched tone (2.5 kHz) signaling the start. Simultaneously, the system reduced the lever's resistance to a low-force setting, allowing the mouse to move it. Upon finishing the trial, successful holds were paired with a high-pitched tone (5 kHz), and failures indicated by a low-pitched tone (1 kHz), all emitted from the buzzer.

The training was divided into two distinct phases:

Stage 1: An introductory phase where any lever pull exceeding 3° prompted the dispensation of a water drop, fostering initial engagement. Additionally, to acquaint mice with the water source, a 'free water drop' was dispensed every 15 minutes upon nose-poking, independent of lever pulling. Advancement to Stage 2 was contingent on the mouse performing a minimum of 100 pulls in one day.

Stage 2: Transitioning to this stage at the commencement of the following day, mice were required to pull and hold the lever within the 6–24° target range. The required 'hold time' started at zero seconds and was adjusted daily at midnight based on the 75th percentile of the previous day's hold times across all trials. The new required hold time was only set if it exceeded the previous value and if the mouse achieved a minimum of 30% successful trials out of at least 100 attempts. Should a mouse's daily success fall below 10% for two consecutive days, the required hold time was manually reset to the last successful benchmark to prevent dehydration and loss of motivation.

The PiPaw2.0 task dynamically adjusted the required hold time based on each mouse's performance to maintain an individualized level of challenge and prevent disengagement due to task monotony or excessive difficulty. Daily trial counts were recorded and monitored over the study period to track engagement. The maximum required hold time was capped at one second, with the overall hold time allowed up to five seconds, beyond which the lever resistance reverted to the high-force state. To ensure adequate hydration, the system calculated the shortfall from the ideal daily intake of 1 mL of water based on the previous day's earned rewards. Any deficit was compensated with 'free water drops' the following day, delivered on the same fixed schedule established in Stage 1, and factored into the total count for the subsequent day's calculations.

### Data analysis

**General procedures.** The preprocessing and analysis of all data were conducted using MATLAB (2021b). To facilitate reproducibility and further research, we have made all scripts and data publicly available on GitHub (https://github.com/ubcbraincircuits/PiPaw2.0_DataAnalysis) and the Federated Research Data Repository (FRDR https://doi.org/10.20383/103.0869) respectively. For statistical tests and the generation of graphical representations, we employed both MATLAB and GraphPad Prism (v8).

Given the variability in the duration for which animals were housed in the cages, we standardized the analysis time frame for daily metrics to the first two months (58 days) of housing to maintain consistency across subjects. This period was selected as it represented a substantial time frame to observe significant behavioral and neural adaptations while accounting for potential variations in individual housing durations. For analyses that did not rely on daily metrics, we utilized data from the entire duration of the animals' stay in the cages, thereby maximizing the use of the available data.

**Multi-level modeling of daily average hold times.** Quantitative analysis via linear mixed-effects modeling (lmer function) was conducted to assess the effects of cage- and mouse-related variables on daily average hold time (DAHT) using the lme4 package in R (Version

4.2.2) with the Nelder-Mead optimizer. Intraclass correlation coefficients (ICC) were obtained via the performance package. The alpha level for all tests was p = 0.05.

First, we determined the inclusion or exclusion of independent variables using their ICCs from null models, i.e., a model of one independent variable against the DAHT. Variables with low ICCs were excluded from the final model. Then, we gradually added effects and interactions of independent variables to the time-only model (i.e., model of only 'days in cage' and DAHT). Deviance testing via ANOVA was conducted following each addition of effects or interactions to assess whether its inclusion improved or worsened the time-only model. Since all independent variables were time-invariant, they were set as fixed effects. DAHT was set as a random effect due to being time-variant.

**Trial hold time probabilistic modeling.** In our study, the analysis of hold time data revealed a bimodal distribution, with one mode near zero and the other near the target hold time. This observation led us to adopt a dual-process analytical approach, inspired by previous methodologies [27,28], and grounded in the concept of probabilistic modeling. We conceptualized the hold time data as emerging from two distinct behavioral processes. The first, a rapid and seemingly impulsive process, was represented as an exponential distribution, and modeled the shorter hold times as outcomes of a Poisson-like process, occurring at a certain rate ($\kappa$). These shorter trials were hypothesized to constitute a proportion 'w' of all the responses recorded. The second process, characterized by more deliberate and timed responses, was modeled using a Gaussian distribution, reflecting the trials where animals held the lever closer to the target duration.

To encapsulate this dual-process behavior, we formulated the overall model for the hold time distribution as follows:

$$p\left(\text{hold time} = x\right) = \left[w \times \text{Exponential}\left(x; \kappa\right)\right] + \left[\left(1-w\right) \times \text{Gaussian}\left(x; \mu, \sigma^2\right)\right]$$

Here, 'w' represents the weight of the exponential distribution in the overall model, '$\kappa$' is the rate parameter for the exponential component, and '$\mu$' and '$\sigma^2$' are the mean and variance of the Gaussian component, respectively.

For parameter optimization and determination of the daily *w* values, we employed MATLAB's '*fminsearch*' function. This approach enabled us to perform individual fits for each day, allowing for a day-by-day analysis of the evolving response patterns and their underlying probabilistic mechanisms.

**Measurement of movement jerkiness.** The raw trajectory data underwent preprocessing to exclude any trials that were shorter than 20 samples (0.05 seconds), as such brief movements were considered non-representative of the intended task dynamics and could introduce noise into the analysis (<5% trials). A high-pass filter was applied to the preprocessed trajectory data using the 'highpass' function in MATLAB, set with a cutoff frequency of 10 Hz. The highpass function utilizes a minimum-order filter with a stopband attenuation of 60 dB and compensates for the delay introduced by the filter, thereby maintaining the integrity of the high-frequency components. The standard deviation of the filtered trajectory served as a quantitative metric of movement jerkiness, reflecting the variability and smoothness of the high-frequency components of the lever movements. For each mouse, these jerkiness values were averaged on a daily basis to assess changes over time and to compare between the WT and zQ175 mice.

**Measurement of trial-to-trial variability.** Dynamic time warping (DTW): To evaluate the dissimilarity between consecutive trial trajectories, we employed DTW using the 'dtw' function in MATLAB, which is a robust method for measuring the similarity between two temporal sequences, even when they vary in length. In our analysis, DTW was applied to pairs

of consecutive trials by stretching the trajectories onto a common set of time instants, in a way that would minimize the sum of Euclidean distances between corresponding points.

**Moving standard deviation:** The trial-to-trial variability was also measured using the moving standard deviation of hold times. This was calculated using the 'movstd' function in MATLAB, which computes the local standard deviation over a sliding window of specified length (5 trials in this analysis).

The DTW distances and moving standard deviation values were normalized by z-scoring over the analyzed trials, as these measures could be affected by trial durations.

**Inter-mouse influence on learning.** For this analysis, the data from all mice within each cage were combined, and the trials were sorted by timestamp to ensure chronological order. The analysis was performed on each possible pair (a "follower", and an "influencer") within a cage; i.e., 6 pairs for a cage of 3 mice. The key parameters for this analysis included a minimum of 10 trials per mouse and a maximum interval of 5 minutes between bouts of activity for the mice within each pair. Each cage's trial data were processed to identify all possible pairs of mice. The analysis focused on comparing the follower's performance before and after the influencer's trial bout. For each pair, we evaluated sequences where one mouse (the "follower"; baseline performance) performed at least 10 trials within a 5-minute window, followed (within 5 minutes or less) by another mouse (the "influencer") completing a similar number of trials within the same time frame. The follower then returned for another bout of trials within 5 minutes of the influencer's activity (see Fig 7A and 7B). The change in success was calculated as $(SR - SR_{Baseline})/ SR_{Baseline}$ where $SR$ and $SR_{Baseline}$ are the success rate of the bout after and before the influencer, respectively.

## Post-task electrophysiological assessment

Following a 10–12 week period in the behavioral testing environment, mice were removed for subsequent electrophysiological analysis. Mice were initially anesthetized using isoflurane and then rapidly decapitated. Brains were promptly excised and hemisected sagittally, isolating the two hemispheres. Sagittal brain slices (250–300 μm) encompassing the dorsal striatum were prepared with a vibratome (Leica Microsystems, VT1000) in chilled artificial cerebrospinal fluid (aCSF). These slices were subsequently relocated to a pre-warmed (37°C) aCSF bath for 30 minutes, followed by maintenance at ambient temperature for over two hours in preparation for extracellular recordings.

The aCSF was composed of 125 mM NaCl, 2.5 mM KCl, 25 mM $NaHCO_3$, 1.25 mM $NaH_2PO_4$, and 10 mM glucose, with different concentrations of MgCl2 and CaCl2: For the slicing process, aCSF contained 0.5 mM $CaCl_2$ and 2.5 mM $MgCl_2$, while all the other aCSF solutions had concentrations of 2 mM $CaCl_2$ and 1 mM $MgCl_2$. The pH of the aCSF was regulated to 7.3–7.4, with an osmolarity maintained at 310 (±3) mOsm/L, and continuously oxygenated with a carbogen mix (95% $O_2$/5% $CO_2$).

In the recording chamber, slices were bathed in aCSF at room temperature, containing picrotoxin (50 μM; Tocris Bioscience) to suppress $GABA_A$ receptor-mediated inhibition. After a stabilization period of at least 20 minutes, field excitatory postsynaptic potentials (fEPSPs) were measured. Recording and stimulating electrodes were positioned within the dorsolateral striatum and 300–500 μm apart as in [37]. Stimuli were delivered every 15 seconds, as a paired pulse 50 ms apart, with intensity calibrated to elicit substantial yet submaximal fEPSPs (0.3–0.6 mV). The response to the first pulse was used for analysis.

Recordings were stabilized for 10 minutes to confirm the consistency of fEPSP amplitudes before the high-frequency stimulation (HFS) application. The HFS protocol consisted of four 100 Hz stimulus trains for one second each, interspersed with 10-second intervals, using the same stimulation parameters as the baseline pulses. Post-HFS fEPSPs were monitored for 40

minutes to evaluate the striatal responses. A 5-minute pre-HFS period served as a baseline for comparison against the 30–35 minute post-HFS interval.

## Supporting information

**S1 Video. Video of six random trials from an expert mouse (6QP01) performing lever pulling in the PiPaw2.0 chamber, temporally aligned to the start of each pull.** (MP4)

**S1 Appendix. Intra-class correlation and multilevel modeling of daily average hold time.** (DOCX)

**S1 Fig. Average weekly weight of WT and zQ175 mice during the PiPaw2.0 testing.** WT mice exhibited a steady increase in body weight over time, while zQ175 mice showed minimal weight gain and maintained a significantly lighter body weight (2–10% lower) compared to WT mice (RM two-way ANOVA, genotype p = 0.116 $F_{(1, 22)}$ = 2.675, weeks p < 0.0001 $F_{(11, 240)}$ = 7.132, interaction p = 0.001 $F_{(11, 240)}$ = 2.797). Data are presented as mean ± SEM. (TIF)

## Acknowledgments

We are grateful to Dr. Lily Zhang for assistance with animal husbandry and genotyping and Selina Park for critically proof-reading and reviewing the manuscript. This project was supported by resources available through the NeuroImaging and NeuroComputation Center at the Djavad Mowafaghian Center for Brain Health (RRID:SCR_019086).

## Author contributions

**Conceptualization:** Daniel Ramandi, Marja D. Sepers, Brian Han, Cameron L. Woodard, Timothy H. Murphy, Lynn A. Raymond.

**Data curation:** Daniel Ramandi, Marja D. Sepers, Zefang Wang, Brian Han, Timothy H. Murphy, Lynn A. Raymond.

**Formal analysis:** Daniel Ramandi, Marja D. Sepers, Zefang Wang, Brian Han.

**Funding acquisition:** Timothy H. Murphy, Lynn A. Raymond.

**Investigation:** Daniel Ramandi.

**Methodology:** Daniel Ramandi, Marja D. Sepers, Cameron L. Woodard, Timothy H. Murphy, Lynn A. Raymond.

**Resources:** Timothy H. Murphy, Lynn A. Raymond.

**Software:** Daniel Ramandi, Brian Han, Cameron L. Woodard.

**Supervision:** Timothy H. Murphy, Lynn A. Raymond.

**Writing – original draft:** Daniel Ramandi, Lynn A. Raymond.

**Writing – review & editing:** Daniel Ramandi, Marja D. Sepers, Zefang Wang, Brian Han, Cameron L. Woodard, Timothy H. Murphy, Lynn A. Raymond.

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
