## [Decision Letter · Decision Letter 0]

21 Nov 2024

PONE-D-24-48341Home cage-based insights into motor learning and strategy adaptation in a Huntington Disease mouse modelPLOS ONE

Dear Dr. Raymond,

Thank you for submitting your manuscript to PLOS ONE. After careful consideration, we feel that it has merit but does not fully meet PLOS ONE’s publication criteria as it currently stands. Therefore, we invite you to submit a revised version of the manuscript that addresses the points raised during the review process.

We look forward to receiving your revised manuscript.

Kind regards,

Miriam A Hickey, PhD

Academic Editor

PLOS ONE

Journal requirements:    When submitting your revision, we need you to address these additional requirements. 1. Please ensure that your manuscript meets PLOS ONE's style requirements, including those for file naming. The PLOS ONE style templates can be found at https://journals.plos.org/plosone/s/file?id=wjVg/PLOSOne_formatting_sample_main_body.pdf and https://journals.plos.org/plosone/s/file?id=ba62/PLOSOne_formatting_sample_title_authors_affiliations.pdf 2. Please amend your list of authors on the manuscript to ensure that each author is linked to an affiliation. Authors’ affiliations should reflect the institution where the work was done (if authors moved subsequently, you can also list the new affiliation stating “current affiliation:….” as necessary). 3. Please include a caption for figure 1. 4. Thank you for stating the following financial disclosure:  [Funding was provided by the Canadian Institutes of Health Research PJT-178043 and Natural Sciences and Engineering Research Council AWD-021238 to L.A.R, Canadian Institutes of Health Research foundation grant FDN-143209 to T.H.M, University of British Columbia Faculty of Medicine Graduate Award #6442 to D.R, and Canadian Institutes of Health Research Canada Graduate Scholarship-Doctoral to C.L.W. ].  Please state what role the funders took in the study.  If the funders had no role, please state: ""The funders had no role in study design, data collection and analysis, decision to publish, or preparation of the manuscript."" If this statement is not correct you must amend it as needed. Please include this amended Role of Funder statement in your cover letter; we will change the online submission form on your behalf. 5. Please note that your Data Availability Statement is currently missing [the repository name and/or the DOI/accession number of each dataset OR a direct link to access each database]. If your manuscript is accepted for publication, you will be asked to provide these details on a very short timeline. We therefore suggest that you provide this information now, though we will not hold up the peer review process if you are unable. 6. Please include captions for your Supporting Information files at the end of your manuscript, and update any in-text citations to match accordingly. Please see our Supporting Information guidelines for more information: http://journals.plos.org/plosone/s/supporting-information. 

Additional Editor Comments:

Please provide F statistics with numerator and denominator for all ANOVAs.

For all graphs, please define error bars.

Figure 3

Please discuss this in a little more detail - Fig 3C shows a quite large reduction in variability in the zQ175 group at 26d approx - what could underlie this?

Fig 3E Are these data from all trials? Have the authors examined at e.g., day 56 versus day 21 and day 1?

Figure 3E - X axis, please consider providing a more explanatory title, e.g., (number of consecutive trials within a single entry)

Methods

To adhere to ARRIVE guidelines

--please briefly discuss the group sizes used (e.g., sample size calculations and power analyses).

--please briefly discuss housing (were mice of different genotypes housed together)

Reviewers' comments:

Reviewer's Responses to Questions

**Comments to the Author**

1. Is the manuscript technically sound, and do the data support the conclusions?

Reviewer #1: Yes

Reviewer #2: Partly

2. Has the statistical analysis been performed appropriately and rigorously? 

Reviewer #1: Yes

Reviewer #2: Yes

3. Have the authors made all data underlying the findings in their manuscript fully available?

Reviewer #1: Yes

Reviewer #2: Yes

4. Is the manuscript presented in an intelligible fashion and written in standard English?

Reviewer #1: Yes

Reviewer #2: Yes

5. Review Comments to the Author

Reviewer #1: Preclinical research into neurodegenerative diseases, including Huntington’s Disease, depends in part upon motor, cognitive, and social assays performed in disease-modeling animals. However, laboratory animals are highly sensitive to environmental changes and individual handlers, making it challenging to execute these experiments and to interpret results. Automated behavior tasks performed in the home cage can mitigate those challenges, allowing animals to behave in a more natural, unstressed manner, without an observer present.

Here, the authors convincingly demonstrate the feasibility and value of a novel home-cage behavior monitoring apparatus requiring subject mice to press a lever to receive drinking water. As the animals master the skill, the duration for which they must hold the lever increases, necessitating a flexible problem-solving strategy on the part of the animals. The zQ175 knock-in Huntington’s Disease model mice acquired the initial motor skill similarly to their wild type littermates, but struggled to adapt to the progressive hold-challenge, resulting in fewer, shorter hold bouts. The authors also found that wild type mice, but not HD mice, had more successful trials if they followed a successful “good influencer,” suggesting a possible social learning deficit as part of the mouse’s HD symptomology. Finally, the authors found an increase in fEPSPs in the left hemisphere of wild type mice who had completed this hold-reward experiment compared to the same mouse’s right hemisphere, indicating an increase in plasticity contralateral to the gripping paw. No such difference in lateralization was observed in the HD mice.

This home-cage system has several compelling advantages. In particular, it allows for social housing, which is profoundly important for animal wellbeing as well as for eliciting the most naturalistic behavior results. This is also helped by the trials being run in the home cage, a familiar and comfortable environment. The RFID scanner detects individuals entering the trial box and presents them with a lever-pull challenge customized to their level of ability, allowing co-housed animals to progress at different rates. This reviewer is satisfied that the authors have a good system in place to minimize the effects of thirst itself on animal health. The tone played for a successful trial allowed the researchers to observe subtle and very interesting social behaviors in animals. This is especially interesting, as few tests exist to measure the social transmission of knowledge among mice.

There is a punctuation error on line 235, and I encourage the authors to triple-check their punctuation around parentheses. This comment aside, this reviewer feels the authors have demonstrated an interesting and valuable method of measuring motor learning and cognition with the ability to detect learning and coordination deficits in mouse models of neurodegeneration. This method should be of particular interest to the field as it allows for the dissection of subtle social behaviors that would be difficult to capture outside of the mouse’s home environment. This reviewer happily recommends this manuscript for publication without the need for major revisions. The author’s are to be congratulated for a well conducted study and excellently prepared manuscript.

Reviewer #2: Many thanks for the opportunity to review this manuscript. The study provides interesting data on assessing a specific pattern of upper limb motor function control in mice, using a home cage lever-pulling device. Furthermore, the study shows some comparative data collected by this device in zQ175 mice as a model of HD and WT mice- showing some valuable differences on the capacity to readjust to specific pulling task that the authors claimed may be associated with the onset of HD phenotype as also assess further by changes on electrophysiological patterns on the striatum. As such, the study is interesting, demonstration further the benefit of such home based and more naturalistic approaches to assess motor dysfunction in mouse models.

Overall I read the manuscript with interest and I agree with the authors comments on the value of continue developing such technical approaches to study more naturalistic behaviour in prey species like the lab mouse. However, I do think the study has some major challenges once we thoroughly analysed the data and the conclusions that the authors draw form the data. Firstly, the main aim of the study is to validate a new/ revised operant task system, PiPaw2.0 to assess deficits on voluntary movement on these zQ175 HD mice- but the authors already published some early work using similar approaches on a PiPaw system-using the same mice (see reference Woodard CL, Sepers MD, Raymond LA. Impaired Refinement of Kinematic Variability in 821 Huntington Disease Mice on an Automated Home Cage Forelimb Motor Task. J Neurosci.822 2021 Oct 13;41(41):8589–602. While the authors indeed, are now presenting further data- there is no direct comparative analysis on the data acquired by using initial PiPaw system and importantly, how the new 2.0 system is improving the outcomes- note also that looking at the results, many of the data is non-significant which could also query the sensitivity of the system to detect suttled changes. Yes, the reviewer still believes that the data provided in the this paper is of interest. Secondly, there is little information on the study plan and the frequency of study in individual animals- as some of this tasks may be associated with fatigue and exhaustion, it would be good to see a clear experimental design pattern supported by physiological data such as changes on body weight between the animals and groups, any changes on body conditions, any signs of distress on the animals-that could also be an important confounding factor on the overall performance ability of an individual animal. Thirdly, while the authors do a good job on defining many behaviour parameters and their analysis- sometimes it feels like there is an overarching assumption between a specific behaviour and the biological relevance of such functional response. The study mostly provides functional data-with very limited biological data and/or any other type of biomarker data that could aligned the functional data with some clear changes on the brain- atm, this is lacking. There is some informative data on electrophys with some LTP potentiation data- but is this enough to fully associated such changes on the striatum with a HD phenotype on these animals? I would recommend the authors to :

-provide further clarity and experimental data on how this system improved and goes further beyond to the earlier piPaw data (group study in 2021) as this is described as main objective

- provides some further physiological/phenotype data on the animals studies including BW, body conditions, pain. distress along with other HD biomarkers of HD disease- I agree that some of these may be challenging some support on biological data would be very useful. Also provide further clarity of cumulative effects on animals during testing to understand the dynamics of study and effects on fatigue.

-further biomarker/ biological qualification on the WT and HD mouse models so that functional changes are better studies with some biological support. The authors may have some paralel data on the HD mice on what would be their disease phenotype at 6-7 months.

The results showed many parameters, and many of them of valuable relevance but sometimes there is a far too direct pressumption on who such changes may be linked to the onset /progress of the HD phenotype on the studies mice (QZ174) -so further biological data should be provided to confirm such hypothetized causative effects. This is important to provide further scientific robustness to the fucnctional data- as otherwise the study may be better aligned to a development technical approach with restricted predictability value on the HD models.

There are some other minor issues such as:

a) any differences between sexes?

b? why 6-7months were selected on the study and what would be the differences in body weight curves between the HA and WT line?

c) what are the WT mice-was is the background for the qZ175 mice?

d) please provide info on bedding and cage sizes vs group density per cage as this may affect social interaction and the effects of inter-mouse effects

e)water supply? how was this regulated for the operant conditioning test?

f) did animals ever reac a 20% body weight loss- this would be considered a severe level of distress so how would you envisage such animals to perform in a non-clinical illness stage and show naturalistic functional outcomes- any animals with a ill-associated homeostasis is very likely that will impact on functional assessments?-how much of the effects detected are seen by such critical physiological status vs the onset/progress of HD phenotype?

g) please provide a better outline methodology on cumulative effects on the animals studied so there is better information associated to risk for fatigue/ exhaustion/boredomness/ motivation.

Many thanks again for the opportunity to revise this work; as abovementioned, it is of valuable interest but needs further data support and certain clarity in some methodological aspects to strengthen the predictive impact of the functional data provided.

6. PLOS authors have the option to publish the peer review history of their article (what does this mean? ). If published, this will include your full peer review and any attached files.

**Do you want your identity to be public for this peer review?** For information about this choice, including consent withdrawal, please see our Privacy Policy .

Reviewer #1: No

Reviewer #2: No

---

## [Author Response · Author response to Decision Letter 1]

5 Jan 2025

We sincerely thank you and the reviewers for the time and effort spent providing thoughtful feedback on our manuscript. We greatly appreciate the constructive comments and suggestions, which have helped us improve the clarity, robustness, and overall quality of our work. We have made the requested revisions and carefully addressed each point raised by the reviewers.

Our detailed responses to each comment, along with the corresponding edits to the manuscript, are outlined below.

We hope that our revisions meet the expectations of the reviewers and the editorial team, and we look forward to your feedback.

Editor Comments:

Comment: Please provide F statistics with numerator and denominator for all ANOVAs.

Response: Thank you for your feedback. We have added the F statistics, including numerator and denominator degrees of freedom, for all ANOVAs in the relevant figure captions where p-values are reported.

Comment: For all graphs, please define error bars.

Response: Thank you for your comment. We have defined the error bars in all figure captions as representing the mean ± SEM.

Figure 3:

Comment: Please discuss this in a little more detail - Fig 3C shows a quite large reduction in variability in the zQ175 group at 26d approx - what could underlie this?

Response: Thank you for pointing out the notable reduction in variability in the zQ175 group around day 26 in Fig. 3C. Upon closer inspection, this reduction appears to be driven by decreased variability in the performance of individual animals rather than an overall biological shift. During the earlier stages of learning, there are larger fluctuations in the number of trials per entrance, likely reflecting a more variable approach in behavior as the mice adapt to the task. By around day 26, these fluctuations begin to stabilize, as indicated by the smaller error bars, suggesting that the mice may have reached a more consistent engagement pattern.

We opted not to speculate further on potential biological mechanisms underlying this reduction, as we currently lack direct evidence to support specific hypotheses. We have, however, added a brief note in that section to acknowledge this trend:

The observed reduction in variability appears to arise from a decrease in variability between individual animals rather than a sudden behavioral shift. This stabilization likely reflects a gradual settling into more consistent engagement patterns across the group, rather than an abrupt change in task performance dynamics.

Comment: Fig 3E Are these data from all trials? Have the authors examined at e.g., day 56 versus day 21 and day 1?

Response: Thank you for your question. The data in Fig. 3E represents all trials across the entire testing period. Analyzing the bout size distribution and success rate on a daily basis results in highly variable distributions due to the limited number of data points per day. Instead, we have examined multiple time windows—specifically early (week 1), middle (e.g., week 2-3), and late (week 7-8)—and observed consistent trends in bout size distribution and success rates. The genotype-dependent differences became more pronounced during the middle and late stages, which aligns with the divergence in the number of trials per entrance shown in Fig. 3C.

We believe that since these patterns can already be inferred from Fig. 3C, and would not change the conclusion, adding further data in the results section would introduce redundancy.

Comment: Figure 3E - X axis, please consider providing a more explanatory title, e.g., (number of consecutive trials within a single entry)

Response: Thank you for the suggestion. We have updated the X-axis label in Fig. 3E to "Bout Size (Number of Trials Per Entrance)" to provide a more explanatory and intuitive description.

Methods

To adhere to ARRIVE guidelines

Comment: --please briefly discuss the group sizes used (e.g., sample size calculations and power analyses).

Response: In our study, we used 10 zQ175 mice and 14 WT mice, consistent with sample sizes commonly used in behavioral studies involving operant conditioning and motor learning tasks.

While a formal power analysis was not performed prior to the study, our group size was informed by prior studies that detected significant motor learning differences in HD models using similar automated home-cage paradigms. Moreover, we applied statistical methods such as multilevel modeling, which are robust to varying sample sizes and account for both within- and between-group variability, increasing the reliability of our results.

We added the following to the methods section:

Our group size was informed by prior studies that detected significant motor learning differences in HD models using similar automated home-cage paradigms.

Comment: --please briefly discuss housing (were mice of different genotypes housed together)

Response: In this study littermates were housed together, regardless of the genotype. The following was added to the methods section to clarify this:

Mice were housed at a density of 2 or 3 mice per cage, and including either or both genotypes, as genotypes were not separated.

Reviewer #1: Preclinical research into neurodegenerative diseases, including Huntington’s Disease, depends in part upon motor, cognitive, and social assays performed in disease-modeling animals. However, laboratory animals are highly sensitive to environmental changes and individual handlers, making it challenging to execute these experiments and to interpret results. Automated behavior tasks performed in the home cage can mitigate those challenges, allowing animals to behave in a more natural, unstressed manner, without an observer present.

Here, the authors convincingly demonstrate the feasibility and value of a novel home-cage behavior monitoring apparatus requiring subject mice to press a lever to receive drinking water. As the animals master the skill, the duration for which they must hold the lever increases, necessitating a flexible problem-solving strategy on the part of the animals. The zQ175 knock-in Huntington’s Disease model mice acquired the initial motor skill similarly to their wild type littermates, but struggled to adapt to the progressive hold-challenge, resulting in fewer, shorter hold bouts. The authors also found that wild type mice, but not HD mice, had more successful trials if they followed a successful “good influencer,” suggesting a possible social learning deficit as part of the mouse’s HD symptomology. Finally, the authors found an increase in fEPSPs in the left hemisphere of wild type mice who had completed this hold-reward experiment compared to the same mouse’s right hemisphere, indicating an increase in plasticity contralateral to the gripping paw. No such difference in lateralization was observed in the HD mice.

This home-cage system has several compelling advantages. In particular, it allows for social housing, which is profoundly important for animal wellbeing as well as for eliciting the most naturalistic behavior results. This is also helped by the trials being run in the home cage, a familiar and comfortable environment. The RFID scanner detects individuals entering the trial box and presents them with a lever-pull challenge customized to their level of ability, allowing co-housed animals to progress at different rates. This reviewer is satisfied that the authors have a good system in place to minimize the effects of thirst itself on animal health. The tone played for a successful trial allowed the researchers to observe subtle and very interesting social behaviors in animals. This is especially interesting, as few tests exist to measure the social transmission of knowledge among mice.

There is a punctuation error on line 235, and I encourage the authors to triple-check their punctuation around parentheses. This comment aside, this reviewer feels the authors have demonstrated an interesting and valuable method of measuring motor learning and cognition with the ability to detect learning and coordination deficits in mouse models of neurodegeneration. This method should be of particular interest to the field as it allows for the dissection of subtle social behaviors that would be difficult to capture outside of the mouse’s home environment. This reviewer happily recommends this manuscript for publication without the need for major revisions. The author’s are to be congratulated for a well conducted study and excellently prepared manuscript.

Response: We sincerely thank the reviewer for their detailed and thoughtful review, as well as for their kind words of support and appreciation for our work. We are grateful that the reviewer recognizes the value of our home-cage system and its contributions to the study of motor learning, cognitive adaptation, and social interactions in mouse models of neurodegeneration.

We have carefully reviewed the manuscript once more to ensure that all punctuation, particularly around parentheses, is correct. We appreciate the reviewer bringing this to our attention.

Reviewer #2: Many thanks for the opportunity to review this manuscript. The study provides interesting data on assessing a specific pattern of upper limb motor function control in mice, using a home cage lever-pulling device. Furthermore, the study shows some comparative data collected by this device in zQ175 mice as a model of HD and WT mice- showing some valuable differences on the capacity to readjust to specific pulling task that the authors claimed may be associated with the onset of HD phenotype as also assess further by changes on electrophysiological patterns on the striatum. As such, the study is interesting, demonstration further the benefit of such home based and more naturalistic approaches to assess motor dysfunction in mouse models.

Overall I read the manuscript with interest and I agree with the authors comments on the value of continue developing such technical approaches to study more naturalistic behaviour in prey species like the lab mouse. However, I do think the study has some major challenges once we thoroughly analysed the data and the conclusions that the authors draw form the data. Firstly, the main aim of the study is to validate a new/ revised operant task system, PiPaw2.0 to assess deficits on voluntary movement on these zQ175 HD mice- but the authors already published some early work using similar approaches on a PiPaw system-using the same mice (see reference Woodard CL, Sepers MD, Raymond LA. Impaired Refinement of Kinematic Variability in 821 Huntington Disease Mice on an Automated Home Cage Forelimb Motor Task. J Neurosci.822 2021 Oct 13;41(41):8589–602. While the authors indeed, are now presenting further data- there is no direct comparative analysis on the data acquired by using initial PiPaw system and importantly, how the new 2.0 system is improving the outcomes- note also that looking at the results, many of the data is non-significant which could also query the sensitivity of the system to detect suttled changes. Yes, the reviewer still believes that the data provided in the this paper is of interest. Secondly, there is little information on the study plan and the frequency of study in individual animals- as some of this tasks may be associated with fatigue and exhaustion, it would be good to see a clear experimental design pattern supported by physiological data such as changes on body weight between the animals and groups, any changes on body conditions, any signs of distress on the animals-that could also be an important confounding factor on the overall performance ability of an individual animal. Thirdly, while the authors do a good job on defining many behaviour parameters and their analysis- sometimes it feels like there is an overarching assumption between a specific behaviour and the biological relevance of such functional response. The study mostly provides functional data-with very limited biological data and/or any other type of biomarker data that could aligned the functional data with some clear changes on the brain- atm, this is lacking. There is some informative data on electrophys with some LTP potentiation data- but is this enough to fully associated such changes on the striatum with a HD phenotype on these animals? I would recommend the authors to :

Comment: -provide further clarity and experimental data on how this system improved and goes further beyond to the earlier piPaw data (group study in 2021) as this is described as main objective

Response: We sincerely thank the reviewer for their detailed feedback and thoughtful comments. We appreciate the recognition of the study’s novelty and the potential value of naturalistic home-cage-based assessments for studying motor dysfunction in HD mouse models. Below, we address the specific concern regarding the comparison of PiPaw2.0 to our previously published system.

The main differences between our current study and the previous work by Woodard et al. (2021) lie in the HD mouse model used and the task design. The previous study employed symptomatic Q175FDN HD-model mice (Southwell AL et al. Hum Mol Gen 2016: doi: 10.1093/hmg/ddw212); these mice are symptomatic by 9 months of age and were tested on the original PiPaw task at 10 -11m performing a task requiring lever pulls within a narrow goal range, without needing to hold the lever for a specified period of time (original PiPaw task). In contrast, our study focuses on early-stage HD using zQ175 knock-in mice at six months of age, a critical time point for detecting subtle impairments in motor learning.

In PiPaw2.0, we introduced a wider goal range and a dynamic hold time requirement, where the necessary hold duration adapts to the mouse's performance. This change enhances task complexity and ensures the difficulty is personalized, encouraging continued engagement and learning over months. Consequently, the average daily number of trials increased, reflecting sustained engagement.

Furthermore, in PiPaw2.0, the software has been extensively refined to eliminate errors and prevent system failures. The updated codebase supports more robust, uninterrupted data collection. Additionally, our annotated dataset is openly accessible, fostering transparency and enabling reproducibility in behavioral modeling studies.

We believe that providing excessive detail on the technical differences between PiPaw and PiPaw2.0 within the main text may risk confusing the reader and detracting from the broader focus of the study. Instead, we have added the following concise comparison in the introduction section to clarify the key advancements without blurring the main narrative.

This study extends previous work on the PiPaw system by introducing significant changes to the task design and testing a different HD mouse model. Unlike the earlier version, which used a narrow-range lever-pulling task, PiPaw2.0 employs an adaptive hold-time task where the required hold duration of the lever dynamically adjusts based on each mouse’s performance, promoting sustained engagement and continuous learning over months. The updated system also features a more robust codebase, eliminating software errors and enabling reliable long-term data collection, with all annotated data publicly accessible for further analysis.

Comment: a - provides some further physiological/phenotype data on the animals studies including BW, body conditions, pain. distress along with other HD biomarkers of HD disease- I agree that some of these may be challenging some support on biological data would be very useful. Also provide further clarity of cumulative effects on animals during testing to understand the dynamics of study and effects on fatigue.

b -further biomarker/ biological qualification on the WT and HD mouse models so that functional changes are better studies with some biological support. The authors may have some paralel data on the HD mice on what would be their disease phenotype at 6-7 months.

Response: We appreciate the reviewer’s thoughtful suggestion to include additional physiological and biological data to support the functional findings in our study. Below, we address these points in detail:

Physiological Monitoring: As described in the methods section, animal body weights were measured three times per week, and all animals were closely

---

## [Decision Letter · Decision Letter 1]

21 Jan 2025

Home cage-based insights into motor learning and strategy adaptation in a Huntington Disease mouse model

PONE-D-24-48341R1

Dear Dr. Raymond,

We’re pleased to inform you that your manuscript has been judged scientifically suitable for publication and will be formally accepted for publication once it meets all outstanding technical requirements.

Kind regards,

Miriam Ann Hickey, PhD

Academic Editor

PLOS ONE

Additional Editor Comments (optional):

All Editor and Reviewer comments have now been addressed.

Reviewers' comments:

Reviewer's Responses to Questions

**Comments to the Author**

1. If the authors have adequately addressed your comments raised in a previous round of review and you feel that this manuscript is now acceptable for publication, you may indicate that here to bypass the “Comments to the Author” section, enter your conflict of interest statement in the “Confidential to Editor” section, and submit your "Accept" recommendation.

Reviewer #2: All comments have been addressed

2. Is the manuscript technically sound, and do the data support the conclusions?

Reviewer #2: Yes

3. Has the statistical analysis been performed appropriately and rigorously? 

Reviewer #2: Yes

4. Have the authors made all data underlying the findings in their manuscript fully available?

Reviewer #2: Yes

5. Is the manuscript presented in an intelligible fashion and written in standard English?

Reviewer #2: Yes

6. Review Comments to the Author

Reviewer #2: Many thanks for the detail answers to my comments and suggestions- good to see the revised manuscript with the addition of some physiological data and further clarifications on the text and methodology. I am happy with the revised version. Just a note that a 20% body weight would be categorised as severe procedure and most likely requiring a HEP. But i do understand that this may required some broader perspectives on the specific of the strain and some local legislation. Thanks for providing further details on this.

7. PLOS authors have the option to publish the peer review history of their article (what does this mean? ). If published, this will include your full peer review and any attached files.

**Do you want your identity to be public for this peer review?** For information about this choice, including consent withdrawal, please see our Privacy Policy .

Reviewer #2: No

---

## [Editor Report · Acceptance letter]

PONE-D-24-48341R1

PLOS ONE

Dear Dr. Raymond,

I'm pleased to inform you that your manuscript has been deemed suitable for publication in PLOS ONE. Congratulations! Your manuscript is now being handed over to our production team.

Kind regards,

on behalf of

Dr. Miriam Ann Hickey

Academic Editor

PLOS ONE